# In-situ direct seawater electrolysis using floating platform in ocean with uncontrollable wave motion

Tao Liu [1,2,3,10] ✉, Zhiyu Zhao[1,2,4,10], Wenbin Tang[5,6,10], Yi Chen[7], Cheng Lan [1,2,4], Liangyu Zhu[2], Wenchuan Jiang[1,2,4], Yifan Wu [1,2,4], Yunpeng Wang[4,6], Zezhou Yang[2], Dongsheng Yang[4,5], Qijun Wang[8], Lunbo Luo[9], Taisheng Liu [7] ✉ & Heping Xie [1,2,3,4] ✉

Direct hydrogen production from inexhaustible seawater using abundant offshore wind power offers a promising pathway for achieving a sustainable energy industry and fuel economy. Various direct seawater electrolysis methods have been demonstrated to be effective at the laboratory scale. However, larger-scale in situ demonstrations that are completely free of corrosion and side reactions in fluctuating oceans are lacking. Here, fluctuating conditions of the ocean were considered for the first time, and seawater electrolysis in wave motion environment was achieved. We present the successful scaling of a floating seawater electrolysis system that employed wind power in Xinghua Bay and the integration of a 1.2 Nm$^3$ h$^{-1}$-scale pilot system. Stable electrolysis operation was achieved for over 240 h with an electrolytic energy consumption of 5 kWh Nm$^{-3}$ H$_2$ and a high purity (>99.9%) of hydrogen under fluctuating ocean conditions (0~0.9 m wave height, 0~15 m s$^{-1}$ wind speed), which is comparable to that during onshore water electrolysis. The concentration of impurity ions in the electrolyte was low and stable over a long period of time under complex and changing scenarios. We identified the technological challenges and performances of the key system components and examined the future outlook for this emerging technology.

Seawater electrolysis using offshore renewable energy as input offers a promising alternative to the unsustainable utilization of fossil fuels[1-7]. The ocean, which accounts for more than 96% of global water storage, is like a giant hydrogen mine. Moreover, vast offshore wind power resources are difficult to develop and utilize in an economically manner, so seawater electrolysis is becoming increasingly attractive for converting wind power into hydrogen energy.

At present, indirect seawater electrolysis (seawater electrolysis after pre-desalination) has been used in several demonstration projects. In 2017, Japan's Toyota Motor Corporation developed the world's

[1]State Key Laboratory of Intelligent Construction and Healthy Operation and Maintenance of Deep Underground Engineering, Shenzhen University & Sichuan University, Shenzhen 518060, China. [2]Institute of New Energy and Low-Carbon Technology, Sichuan University, Chengdu 610065, China. [3]Guangdong Provincial Key Laboratory of Deep Earth Sciences and Geothermal Energy Exploitation and Utilization, College of Civil and Transportation Engineering, Shenzhen University, Shenzhen 518060, China. [4]Shenzhen Key Laboratory of Deep Engineering Science and Green Energy, Institute of Deep Earth Sciences and Green Energy, Shenzhen University, Shenzhen 518060, China. [5]College of Polymer Science and Engineering, Sichuan University, Chengdu 610065, China. [6]School of Chemical Engineering, Sichuan University, Chengdu 610065, China. [7]Dongfang Electric (Fujian) Innovation Institute Co. Ltd, Fuzhou 350108, China. [8]Dongfang Electric Wind Power Co. Ltd, Deyang 618000, China. [9]Fujian Branch, China Three Gorges Corporation, Fuzhou 350014, China. [10]These authors contributed equally: Tao Liu, Zhiyu Zhao, Wenbin Tang. ✉e-mail: liutao3200023@scu.edu.cn; liuts@dongfang.com; xiehp@scu.edu.cn

first vessel powered from hydrogen fuel, Energy Observer, in which hydrogen was generated from seawater electrolysis[8]. In 2019, Tractebel Overdick proposed a hydrogen generation platform using offshore wind as the energy input[9]. In 2022, Sealhyf was revealed to generate green hydrogen in the sea[10]. The system was equipped with a 1-MW PEM electrolyser capable of producing 400 kg of hydrogen per day. These demonstration projects suggest that the coupling of pre-desalination and conventional electrolysers may be a feasible solution for generating hydrogen in the oceans. However, the use of purification systems to eliminate impurity ions may increase energy consumption and engineering costs, making in situ hydrogen production in the oceans difficult[2,11].

Direct seawater electrolysis eliminates the need for additional water purification processes provides an attractive pathway for hydrogen production in the oceans, but many challenges still exist. The composition of seawater is complex, containing at least 92 elements, many of which can exert a significant negative impact on electrolysis[12–14], e.g. the chlorine oxidation reaction not only competes with the oxygen evolution reaction to reduce hydrogen production efficiency but also releases toxic $Cl_2$, which could cause severe corrosion of electrolysis systems; the deposition and adhesion of insoluble substances such as $Mg(OH)_2$ and $Ca(OH)_2$ generated during electrolysis may weaken the transfer functions of membranes and lead to the rapid deactivation of catalysts. Moreover, the neutral environment and lower ionic conductivity of seawater limit electrolysis[15,16]. More importantly, variations in seawater composition due to various factors (such as geographical location and seasonal changes) pose significant challenges in designing seawater electrolysers[17,18].

Since the concept of direct seawater electrolysis was proposed in the 1970s[19], scientists have developed many strategies involving catalyst engineering[20–29], asymmetric electrolytes[30–38], and hydrophilic membrane-based seawater electrolysis[4,6,39,40] to achieve highly efficient hydrogen generation. Reference[41] developed an anodic catalyst consisting of three-dimensional standing arrays of heterolateral $Ni_3S_2$/$Co_3S_4$ (NiCoS) nanosheets uniformly grown on Ni foam, and a double

electrode electrolyser was used to produce hydrogen from alkaline seawater for 100 h at 2.4 A and 2.08 V. Reference[42] designed an asymmetric feed electrolyser, which operated at 5 A and 2.0 V for 100 h in alkaline seawater with a dry cathode. Reference[43] developed a seawater electrolyser combined with Fe, $P-NiSe_2$ NFs catalyst that could operate for 200 h at 7.2 A and 1.8 V in natural seawater. Although these strategies have demonstrated promising outcomes in laboratory settings in specific cases, there are currently no reports of direct seawater electrolysis conducted in situ. Moreover, the design and implementation of direct seawater electrolysis within marine environment using renewable energy remain largely unexplored.

In our previous work[1], seawater electrolysis driven by water phase-transition migration was proposed. This strategy is realised by applying a hydrophobic porous membrane as a gas-path interface and employing a concentrated KOH solution as a self-dampening electrolyte (SDE). During operation, the water vapour pressure difference between the seawater and the SDE induces spontaneous seawater gasification on the seawater side and the diffusion of water vapour through the porous membrane to the SDE side, where it is reliquified by absorption by the SDE. When the water migration rate equals the electrolysis rate, continuous and stable water migration via a 'liquid–gas–liquid' mechanism is realised to provide fresh water for electrolysis (Fig. 1a). A demo-type seawater electrolyser at the 386 l h$^{-1}$ $H_2$ generation scale was fabricated, and it showed long-term stable operation for over 3200 h at 250 mA cm$^{-2}$ in Shenzhen Bay seawater with a low energy consumption of 5.0 kWh Nm$^{-3}$ $H_2$. This strategy realises efficient size-flexible and scalable direct seawater electrolysis in a way similar to fresh water splitting without a notable increase in operation cost and has high potential for practical application in the ocean. To further validate the viability of large-scale applications integrating renewable energy in the oceans, comprehensive tests on direct seawater electrolysis must be conducted under fluctuating wave conditions.

Here, for the first time, we present a scaled floating platform for in situ direct seawater electrolysis using offshore wind energy and operating at a 1.2 Nm$^3$ h$^{-1}$ $H_2$ generation rate under uncontrolled and

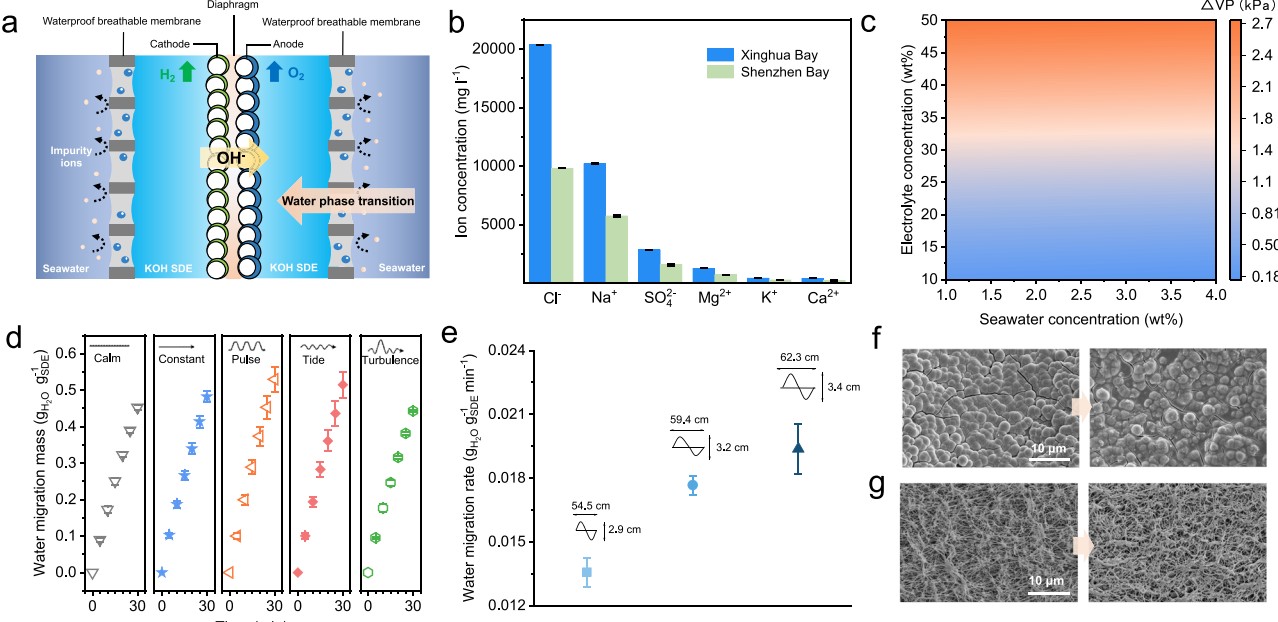

**Fig. 1 | Water migration behaviour in a fluctuating ocean environment.**
**a** Seawater electrolysis driven by a water phase-transition migration mechanism. **b** The ion concentrations in Xinghua Bay and Shenzhen Bay. **c** Water vapour pressure difference at different seawater and SDE concentrations. **d** Water migration mass variation with time under different sea wave modes. Constant mode: stable and continuous output of water flow; pulsed mode: intermittent output of

water flow; tidal mode: water flow output cycling from low to high values; turbulence: random output of water flow. **e** Average water migration rate with different wave heights in the pulsed mode. SEM images of the MoNi/NF anode catalyst (**f**) and PTFE membrane (**g**) before and after 500 h electrolysis. All error bars indicate the standard deviation at three measurements.

fluctuating ocean conditions. First, the influence of the fluctuating environment on the water migration process and on seawater electrolysis was investigated in a laboratory. Then, expanding upon the design of the laboratory-scale demonstration model, three seawater electrolysers were integrated into a system with coupled modules for energy storage, current conversion, $H_2$ detection and transportation modules. This configuration is advantageous because it enables the stable operation of the platform in a fluctuating ocean environment, and wind power is used to supply energy to this type of platform. In the future, there is potential for increasing the economic viability of in situ direct seawater electrolysis based on renewable energy inputs.

## Results

### Process variable correlations during seawater electrolysis

The concentration of seawater in the oceans is a crucial factor influencing electrolysis. Specifically, the composition of seawater varies in different ocean regions (Fig. 1b), leading to distinct vapour pressure differences between seawater and the SDE, thereby influencing the water migration rate and electrolysis balance. To investigate the relationship between seawater and the SDE, a detailed theoretical model was proposed, as outlined in 'Methods'. When the Xinghua Bay seawater concentration approached 2.98 wt% and the KOH solution concentration was 30 wt%, the interfacial water vapour pressure difference was -1.28 kPa, which was lower than the water vapour pressure difference between the Shenzhen Bay seawater (-1.51 wt%) and the SDE (Fig. 1c).

In addition, fluctuations in ocean waves mainly affect the system through their influence on the mass transfer area. The migration rate depends on the instantaneous mass transfer area, which indicates that the water migration mass is an integral function that varies over time. To investigate and compare the difference in water mass transfer in fluctuating environments, we conducted a water migration behaviour test in various simulated sea wave modes (calm, constant, pulsed, tidal and turbulence modes) (Fig. 1d and Supplementary Fig. 1). Compared to those in the calm mode, the water migration masses in the fluctuating environment exhibited slightly greater values. On the one hand, the larger contact area between the moving seawater and the membrane increased the water migration mass per unit time; on the other hand, the movement of seawater could counterbalance the momentary increase in the saline water concentration on the surface of the membrane. The average migration rate of water at the different wave heights and lengths significantly increased (Supplementary Fig. 1b). The water migration rates were -0.0136, 0.0177, and 0.0194 $g_{H2O}\,g^{-1SDE}$ min$^{-1}$ at wave heights of 2.9, 3.2 and 3.4 cm, as well as the corresponding wavelengths of 54.5, 59.4 and 62.3 cm, respectively (Fig. 1e).

Fortunately, dynamic balancing of our strategy could overcome such fluctuations in the oceans. As previously described, the water migration behaviour could be self-regulated by adjusting the water vapour pressure difference with the concentration. For example, if the initial electrolysis rate exceeds the water migration rate, the SDE concentration increases, leading to an increase in the water vapour pressure difference; consequently, the water migration rate also increases to match the electrolysis rate to reach a dynamic balance. In addition, a 500-h durability test of seawater electrolysis in a simulated fluctuating environment was carried out in the laboratory. The SEM images of the clear catalyst and membrane pores provide evidence of the system's adaptability and robustness in such dynamic settings (Fig. 1f, g and Supplementary Fig. 2).

### Floating platform system overview

The kilowatt-scale floating system of in situ direct seawater electrolysis is driven by renewable energy in the oceans (Fig. 2a), and the design is described further in the 'Methods'. An uninterruptible power supply (UPS) containing an energy storage system plays a significant role in regulating stable power to ensure continuous green energy input into the device under intermittent winds (Fig. 2b). A technical illustration of the integrated current conversion, seawater electrolysis, $H_2$ detection and transportation module is shown in Fig. 2c and Supplementary Fig. 3, in which the seawater electrolysis module at the 1.2 Nm$^3$ h$^{-1}$ $H_2$ scale is based on the phase-transition migration strategy previously proposed and comprises three demo-type electrolysers in parallel. Moreover, a condensation device incorporated within the $H_2$ detection module was used to purify the hydrogen gas to obtain the high-purity $H_2$ for subsequent transportation and utilization. Each condensation device receives approximately 700-900 ml of condensed solution per day, which can be further replenished into the electrolysers.

The seawater electrolysis module is a significant part of the floating seawater electrolysis platform, and each electrolyser is immersed in seawater with a maximum water migration area of 13,168 cm$^2$. The water migration interface consists of a hydrophobic polytetrafluoroethylene (PTFE) membrane with gas paths. To allow the electrolyser to adapt to different ocean environments, it is crucial to calculate the relationship between system parameters (such as membrane pores, membrane thickness, SDE concentration, etc.) and the seawater mass transfer rate before operation based on seawater parameters (temperature, concentration, etc.). In addition, a COMSOL simulation was performed to study the distributions of the pressure, concentration, and diffusion rates during operation (Supplementary Fig. 4). As expected, due to the channel design, the electrolyte inside and outside the stack was constantly exchanged, essentially maintaining a 30 wt% concentration. The solution concentration on the electrode surface increased due to the rapid consumption of a large amount of water in the electrolysis process. However, the gas flow also drove the electrolyte exchange process around the stacks, especially oxygen promoting the electrolyte solution flow rate, which is beneficial for the renewal of the electrolyte. In addition, the gas flow of the generated hydrogen gradually increased from bottom to top, which may have occurred to the resistance of the electrolytes, and the hydrogen gas flow drove the renewal and mass transfer of the KOH electrolyte in the cathode cavity. Moreover, due to the accumulation of hydrogen gas and the resistance of the electrolyte solution, the pressure increased from the top (-500 Pa) to the bottom (-3000 Pa). Due to compromises made during implementation at the pilot scale, some of the designs were not optimised, and the nonflowing electrolyser may have resulted in gas pressurization to a certain extent. Nonetheless, the pressure would not damage the diaphragm because the diaphragm can resist higher pressures. The hydrogen generated by 11 electrolytic units flowed smoothly into the confluence pipeline at a rate of 0.2-0.6 m s$^{-1}$. However, there were some design parameters that were worth optimising, e.g. the generated hydrogen that accumulated at the corners of the cavity.

Given the originality and scale of the in situ direct seawater electrolysis platform, notable design and operational challenges had to be overcome, as detailed in 'Methods'. To alleviate fluctuations related to waves, the inclusion of a seawater cabin connected to ocean was proposed so that the waves did not directly impact the electrolysers; instead, the wave fluctuations were partially offset through the floating platform itself (Fig. 2d). This method ensures the application of real-world seawater. Additionally, shaking can be reduced by adding a certain amount of counterweight and anchor (Supplementary Fig. 5). Furthermore, due to salt mist and a high humidity (usually >RH 80%) over the oceans, it was necessary to use a dehumidification system and air circulation system to prevent system corrosion and reduce impurities and a high-humidity atmosphere inside the system (Supplementary Fig. 6).

### Performance under uncontrolled fluctuating ocean conditions

An offshore wind turbine with 10 MW of power located in Xinghua Bay, Fuzhou City, Fujian Province, was one part of a larger wind farm network (Fig. 3a and Supplementary Fig. 7). In this energy network, the operation of the wind turbines is based on optimal energy capture and

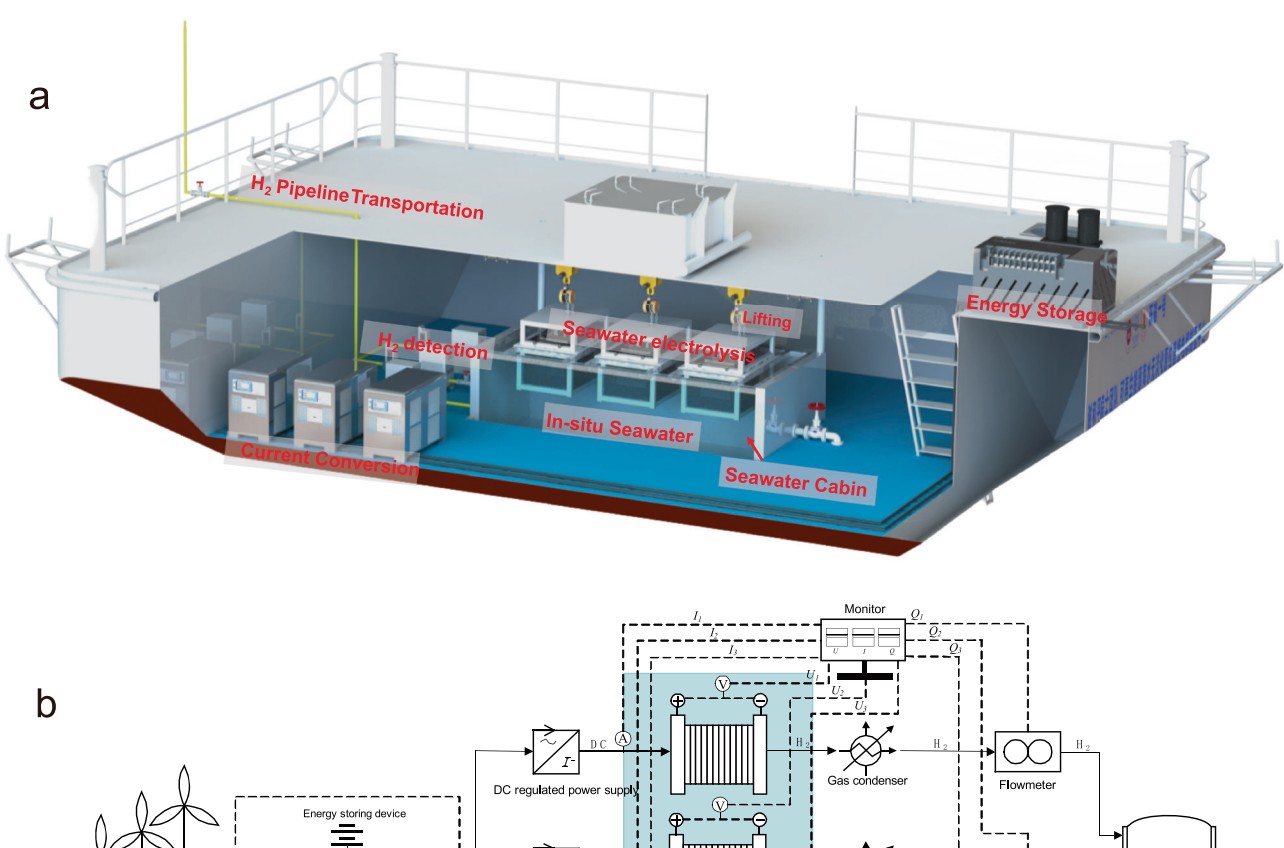

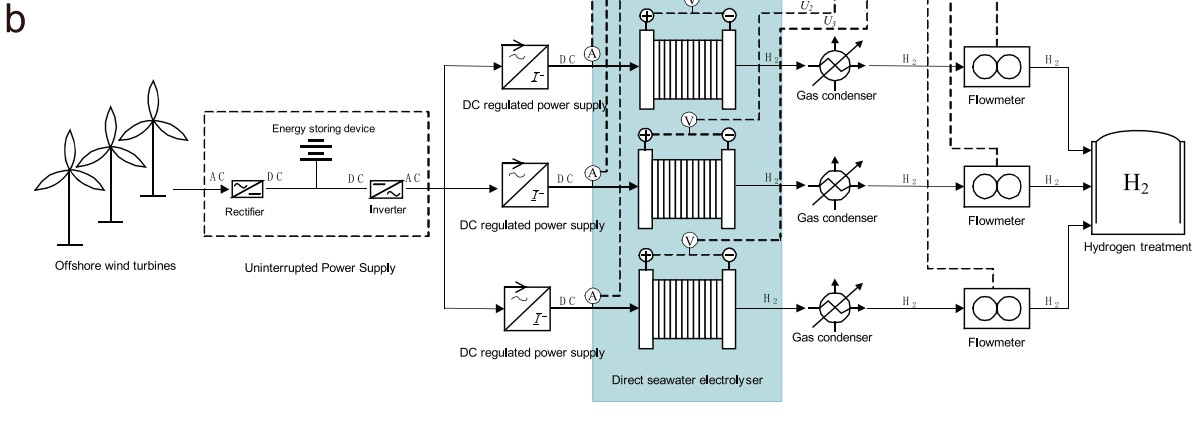

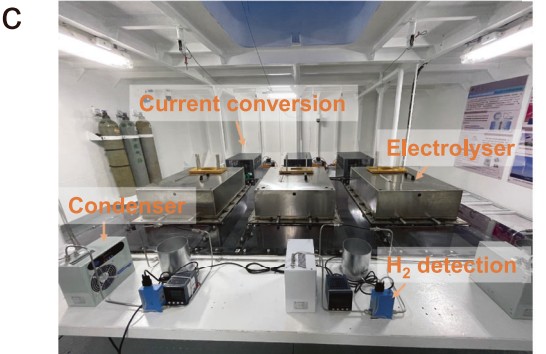

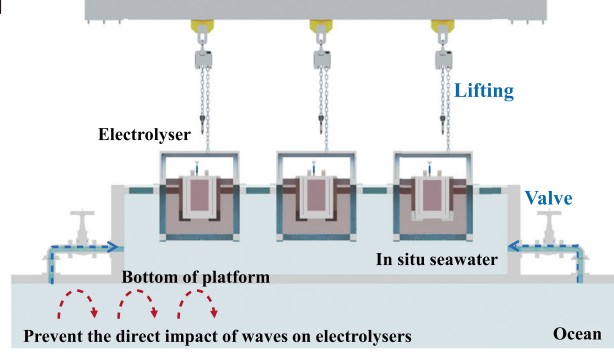

**Fig. 2 | Overview of the platform. a** Illustration of the overall system consisting of the UPS module, current conversion module, seawater electrolysis module, $H_2$ detection module and transportation module. **b** Process diagram of the device ($I$ = current value, $U$ = voltage value, $Q$ = flow value). **c** Photograph of the interior space of the floating platform. **d** Diagram of the seawater cabin connected to the ocean.

power output considerations. Each wind turbine captures and converts wind energy at wind speeds ranging from 3 to 25 m s$^{-1}$ through adjustments to the blade angle and rotation speed of the turbine. Power generation is related to the instantaneous wind speed, as shown in Fig. 3b. Data over a period of 900 min from May 17 to 18 were selected for analysis. During this period, the maximum and minimum

power generation levels of the wind turbine were 10.16 MW and 0.73 MW, respectively. Fluctuations in the wind velocity cause instability in the power output of wind turbines, which is a challenge for the AWE electrolysers due to their lower response rate. Accordingly, a UPS with an energy storage device was used to stabilize the output voltage of the turbine. The UPS could stably output 19.8 kW of

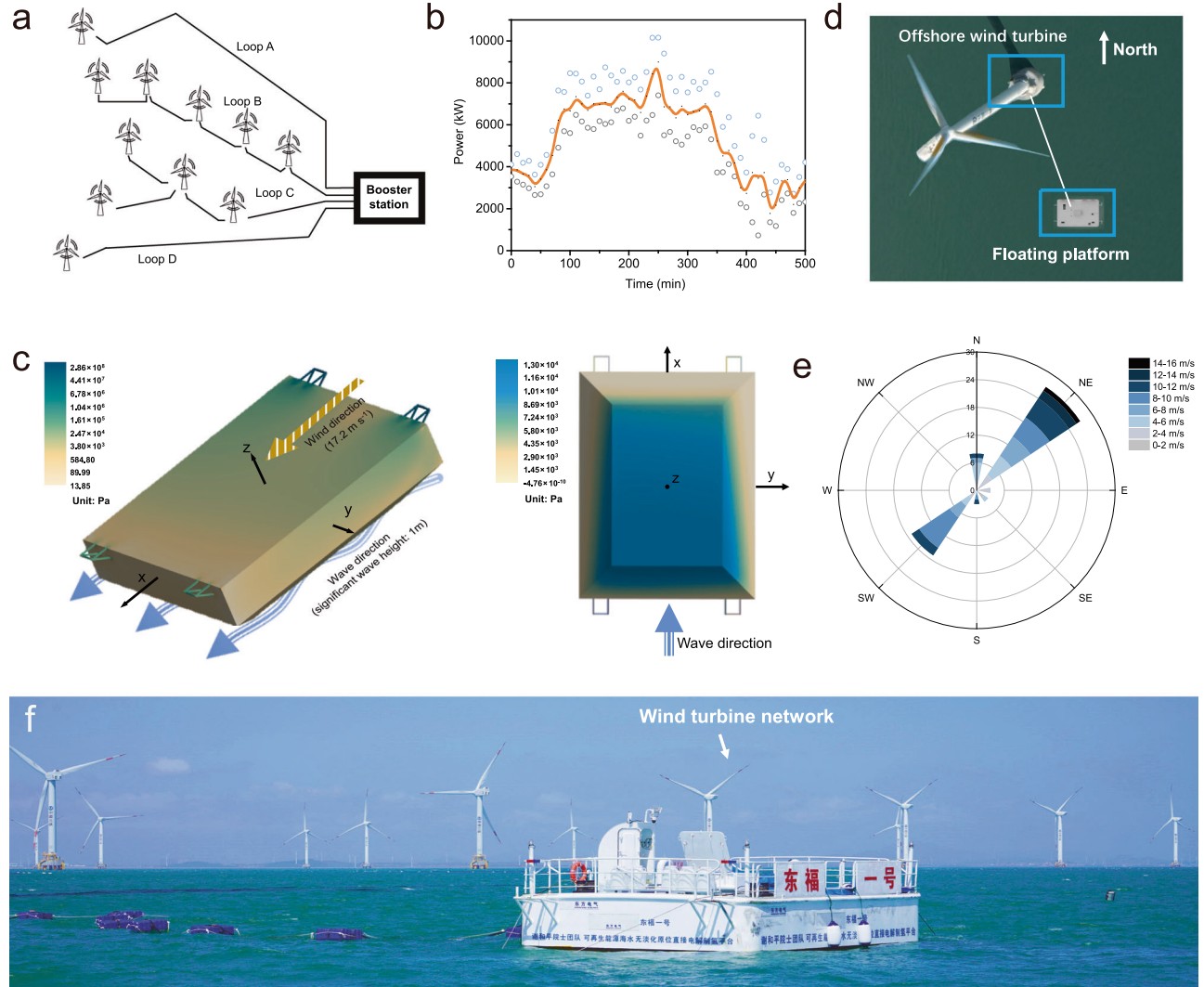

**Fig. 3 | In situ direct seawater electrolysis platform using offshore renewable energy in the ocean. a** Diagram of the wind turbine network. The electricity generated by the wind turbine enters the booster station. When a specific wind turbine cannot generate electricity in a windless environment, the booster station will input energy to this wind turbine for startup and operation. **b** Fluctuation in the turbine power with the wind speed. Within 500 min, the maximum output power of the wind turbine was more than four times that at the lowest level. **c** Pressure and stress distribution of the floating platform under a fluctuating environment. **d** Orientation of the floating platform in the ocean. **e** Directional rose chart of wind during the offshore test. **f** Photograph of the wind turbine network and in-situ direct seawater electrolysis platform in the ocean.

power, providing continuous energy input for the floating seawater electrolysis platform (Supplementary Fig. 8), while excess electricity was stored in the energy storage device. When the offshore wind was insufficient to drive the wind turbine, the energy storage device served as the main power source, providing a stable voltage for the floating platform. This design could ensure the stability of the hydrogen production process.

To ensure the stability and safety of the floating platform in the ocean, the finite element simulation was used to analyse the motion, pressure, and stress distribution of the floating platform under a force 8 wind (-17.2 m s$^{-1}$) and a 1-m significant wave height[44,45]. The floating platform, stabilized by mooring ropes, could swing between 5.5 and 7.9 m along the $X$ direction (sway motion), while it could surge and heave along the $Y$ direction (−0.55-0.56 m) and $Z$ direction (0.54-0.9 m), respectively. The floating platform rotated from −2.5 to 2.0° around the $X$-axis (row), from −8.5 to 7.2° around the $Y$-axis (pitch), and from −12.4 to 10.2° around the $Z$-axis (yaw), which was within a controllable range (Supplementary Fig. 9). In addition, as shown in Fig. 3c, maximum stress occurs at the anchor frame

(-286 MPa), and the maximum fluid pressure on the wet surface of the floating platform body was -13,041 Pa. These values were lower than the yield stress of the ship structural steel employed, ensuring adequate strength for the floating platform.

The floating platform was located approximately southeast of the wind turbine (Fig. 3d), and was connected by 200-m submarine cables (Supplementary Fig. 10). In situ direct seawater electrolysis was performed over a period of 10 days from 17 May to 26 May 2023 at a 1.2 Nm$^3$ h$^{-1}$ H$_2$ generation scale (Fig. 3f). This was the most suitable period for experiments in this ocean area; in other periods, the Maritime Bureau requested evacuation due to typhoon weather conditions. During the 10 days of hydrogen production from seawater, the average temperature of the ambient atmosphere and shallow seawater ranged from -20 to 25 °C. However, the wind speed and wave height varied considerably over time, as shown by the following data: for example, 12:00 AM, May 17, 2023-0.4 m wave height and 7 m s$^{-1}$ wind speed; 0:00 AM, May 18, 2023-0.7 m wave height and 9 m s$^{-1}$ wind speed; 20:00 PM, May 22, 2023-0.9 m wave height and 15 m s$^{-1}$ wind speed. Over the course of one day, the environment conditions

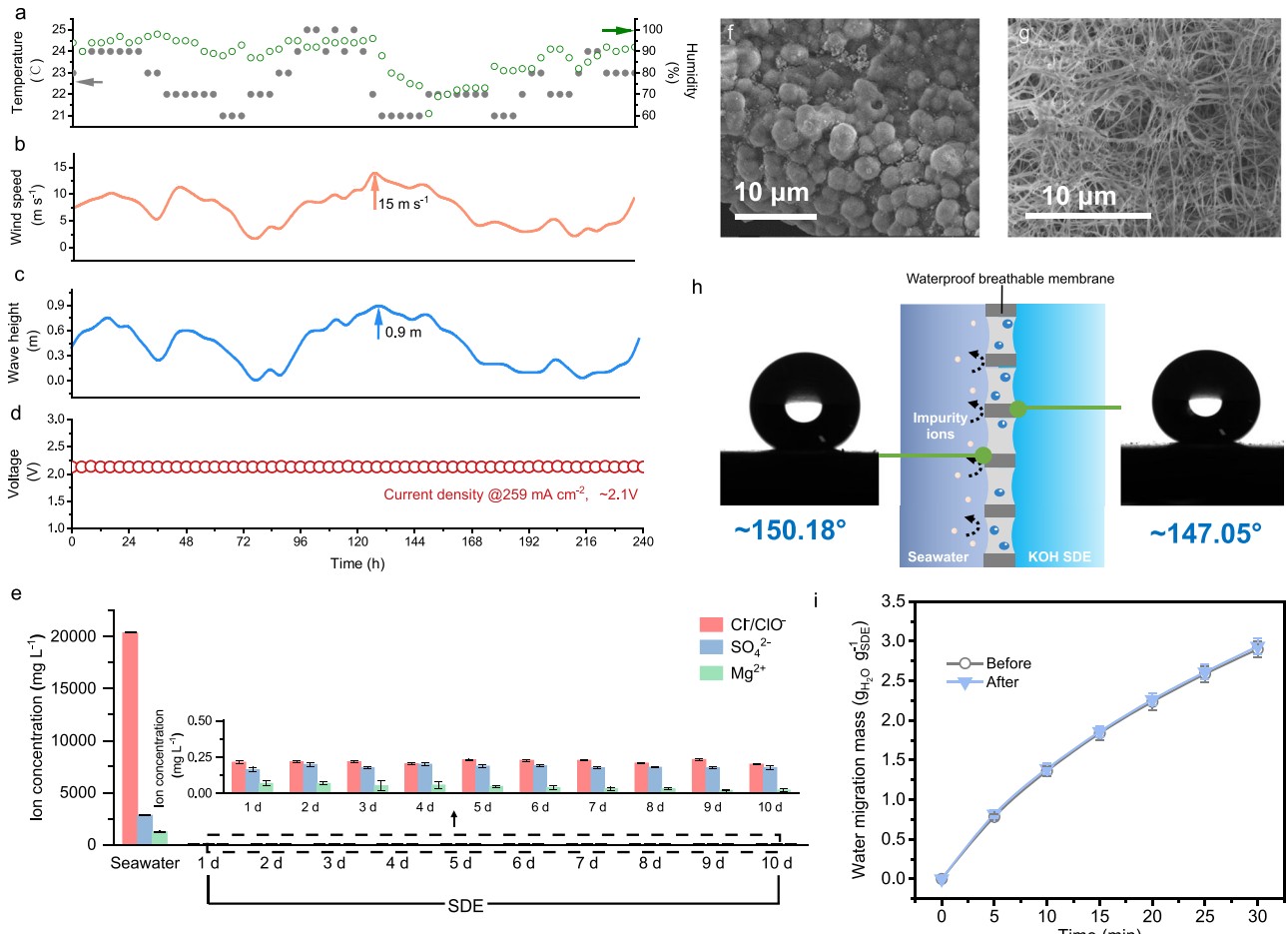

**Fig. 4 | Durability testing in an uncontrolled fluctuating ocean environment.**
**a** Variation in the temperature and relatively humidity in Xinghua Bay. The data
were recorded every 4 h. The wind speed (**b**) and wave height (**c**) during the period
from May 17, 2023 to May 26, 2023. The data were recorded every 4 h. **d** Stability of
the seawater electrolysis system (1#) in an uncontrolled fluctuating ocean
environment. **e** The average ion concentrations of SDE and Xinghua Bay seawater.
SEM images of the catalyst (**f**) and PTFE membrane (**g**) after electrolysis in a fluc-
tuating ocean. **h** Contact angle of the PTFE membrane after electrolysis. **i** Water
migration behaviour of the PTFE membrane before and after 10 days of use. All
error bars indicate the standard deviation at three measurements.

also greatly varied, e.g. at 4:00 AM, May 21, 2023-0 m wave height and
3 m s$^{-1}$ wind speed; at 20:00 PM, May 21, 2023-0.6 m wave height and
8 m s$^{-1}$ wind speed. During this period, the winds were primarily from
the northeast and southwest (Fig. 3e).

At a constant current of 957 A for electrolysis of each electrolyser in
Xinghua Bay seawater and an uncontrolled fluctuating ocean environ-
ment with a 0.3 - 0.9 m wave height and 0-15 m s$^{-1}$ wind speed, the
platform system exhibited good and stable performance for over 240 h
with an energy consumption of ~5.0 kWh Nm$^{-3}$ H$_2$ (Fig. 4 a-d and Sup-
plementary Fig. 11). The voltage of each electrolyser was ~2.14, 2.12 and
2.14 V at 250 mA cm$^{-2}$, which was ~0.1 V higher than the voltage tested in
the laboratory (Supplementary Fig. 12). This may have occurred due to a
slight loosening of the electrode plates and catalysts during transpor-
tation on the road or in the ocean, which led to an increase in the
electronic transmission resistance, but the values were within an
acceptable range. The energy consumption was also similar to that of
onshore industrial alkaline electrolysis using pure water, while the extra
investment of pre-desalination was eliminated. The hydrogen purity
was over 99.9% (Supplementary Fig. 13), satisfying requirements for
commercial use. Moreover, no obvious increase in impurities, such as
Cl$^-$, Mg$^{2+}$, and SO$^{2-}_4$, was detected in the SDE after long-term electrolysis;
the electrolyser exhibited over 99.99% ion-blocking efficiency (Fig. 4e
and Supplementary Fig. 14). The SDE remained clear without precipitate
formation, supporting this point. The above research further illustrated

that the system could withstand certain wave impacts. Scanning
electron microscopy (SEM) analysis revealed that the catalyst layer
maintained its original morphology even after long-term operation
(Fig. 4f). In addition, the OH$^-$ concentration of the KOH SDE was stable
and ranged from 25 to 28%, indicating that the electrolysis and water
migration rates of the system tended to remain consistent at an average
seawater temperature of ~23 °C (Supplementary Fig. 15).

The PTFE membrane was examined after 10 days of operation.
SEM analysis revealed distinct pores in the membrane, suggesting no
notable biofouling or pore blocking (Fig. 4g). The contact angles of the
hydrophobic PTFE membrane ware ~147.05° (with KOH droplets) and
150.18° (with Xinghua Bay seawater droplets) after long-term operation
(Fig. 4h), verifying the suitable environmental tolerance of the multi-
fluorine structure. To confirm that the mass transfer performance of
the PTFE membrane was not affected by seawater waves or impurities,
a water migration behaviour test was further conducted, and the
results were compared to the membrane's performance before the
test. As shown in Fig. 4i, the average water migration in the PTFE
membrane after 10 days of use was ~0.097 g$_{H2O}$ g$^{-1 SDE}$ min$^{-1}$, which was
similar to that with the new membrane (0.096 g$_{H2O}$ g$^{-1 SDE}$ min$^{-1}$).

## Potential and outlook
Eliminating the dependence on pure water during in situ seawater
electrolysis is necessary in the oceans. It may become more applicable in

the oceans if complex processes such as seawater introduction, pre-treatment, and brine treatment can be eliminated. Moreover, reducing energy consumption during electrolysis is important for optimising the cost of hydrogen generation. To improve the performance and economy of our system, the use of a split-type system consisting of a commercial electrolyser and a mass transfer module is proposed to optimised the system (Supplementary Fig. 16). This means that the system could match the performance of the best commercial electrolysers (containing catalysts with higher activity) without changing their structure. In the ocean, complex seawater desalination processes, additional platform space, energy consumption related to pretreatment by desalination, etc., are not required in this mode. This mode of hydrogen production without requirement of pure water and with excellent performance, will have increasingly attractive on in-situ direct seawater electrolysis using offshore wind power as energy input in the ocean.

The scale of offshore wind power is increasing year by year. However, it is difficult to transmit renewable energy power to land due the high associated losses and costs. Many countries are counting on abundant offshore wind power and the flexibility of hydrogen energy to meet energy security and climate goals, and combining these two technologies may provide some advantages. In situ direct seawater electrolysis using wind power as energy input may provide an efficient way to achieve convenient, efficient, and economical energy conversion.

## Discussion

In this study, the influence of wave motion on water migration behaviour and electrolysis processes was first investigated in a laboratory setting to further study seawater electrolysis driven by a water phase-transition migration mechanism in the oceans. It was observed that the wave motion facilitates to enhance the water mass transfer and a certain degree of wave fluctuation could not negatively impact the stability of the core components of the electrolyser (such as membrane or catalyst).

Additionally, for the first time, the operation of a large-scale (6 kW scale of electrolysis, and 1.2 Nm³ h⁻¹ H₂ production) and efficient (>99.9% hydrogen purity) floating platform for in-situ direct seawater electrolysis was successfully demonstrated in the ocean under uncontrolled fluctuating wind and wave conditions. The design and construction of the platform were illustrated, highlighting how critical technological challenges were overcome, e.g. through the utilizing of UPSs and energy storage modules to stabilize electricity or via achieving physical isolation to prevent the impact of waves. In addition, a simulation of the electrolyser and floating platform was conducted to analyse and validate the impact of the process and environmental parameters.

Moreover, the 240 h durability results in Xinghua Bay demonstrated that during system operation in the ocean with offshore wind power as the energy source, the blocking performance (>99.99%) of impurities was good and operation was stable without catalyst corrosion and side reactions, indicating the absolute stable isolation between seawater and SDE in a stronger fluctuation environment (0-0.9 m wave heights, 0-15 m s⁻¹ wind speeds), which is similar to the results in the laboratory. The electrolysis performance (~2.13 V at 250 mA cm⁻²) was slightly higher than that under laboratory conditions since parts of the electrolysers may become loose during transportation, but the results were acceptable. Overall, under specific wind and wave conditions, the system could withstand certain fluctuations and had favourable performance.

For use in the deep seas, the resistance of the floating platform to stronger winds and waves should be considered; moreover, the stability and strength of the hydrophobic porous membrane and electrolysis system still need to be optimised for long-term use under controlled ocean conditions. With the optimisation of the floating platform and the development of offshore wind power, the advantages of in situ direct seawater electrolysis will become increasingly apparent.

## Methods

### Water migration test in the laboratory

Static water migration behaviour was observed according to the weight changes of the SDE electrolytes at 25 °C. Here, KOH and 0.5 M NaCl were considered as the SDE electrolyte and simulated seawater, respectively. A certain quantity of KOH solution and 0.5 M NaCl were placed on both sides of the porous PTFE membrane. The weight changes in the KOH solution were recorded at regular intervals to prepare a curve showing the mass of water migrating over time.

Additionally, an open $60 \times 40 \times 20$ cm³ container was used as a seawater cabin, in which a 0.5 M NaCl solution was used as the simulated seawater, and a wavemaker was used to create different kinds of simulated waves (calm, constant, pulsed, tidal and turbulence modes). A 50 wt% KOH solution as the SDE was placed in a square container and a PTFE membrane with 0.1 μm pore size was lined on the five walls of the container with mass transfer area of 75 cm². Water migration tests were carried out under simulated wave conditions with different wave types and heights. The wave heights of 3.4, 3.2, and 2.9 cm corresponded to wavelengths of 62.32, 59.40, and 54.52 cm, respectively, and the periods were all 1.34 s.

### Seawater electrolysis in a simulated fluctuating environment in the laboratory

To test the durability of the seawater electrolyser in a fluctuating seawater environment, an open $1.4 \times 1.2 \times 0.6$ m³ container containing seawater from Xinghua Bay was used as the seawater cabin, and irregular waves were created to simulate the fluctuating environment of the ocean. The seawater electrolyser was immersed in seawater with a specific water migration area according to the concentration of seawater and SDE. Linear sweep voltammetry (LSV) was conducted using a multimeter. The corresponding voltages between the anode and cathode plates of the electrolyser were obtained at current densities of 0 mA cm⁻², 10 mA cm⁻², 50 mA cm⁻², 100 mA cm⁻², 150 mA cm⁻², 200 mA cm⁻², and 259 mA cm⁻². The electrolysers were operated for more than 500 h at a current of 957 A. The voltage of each electrolyser was recorded as the result of the electrolysis process. After electrolysis, the microstructure of the gas pathways in the PTFE membrane and the morphology of the commercial catalyst were observed via SEM analysis (Thermo Scientific Apreo 2C). The water contact angle of the PTFE membrane was measured at room temperature using a contact angle goniometer (DSA 25, Krüss) with seawater droplets.

### Floating platform design

**UPS module.** A UPS, which contains a lithium-ion battery as an energy storage device, is mainly used to provide an uninterrupted electricity for floating platform. When the wind in the oceans is sufficient to drive a wind turbine, the electricity generated by the wind turbine is stably transported to the platform through the UPS, while the excess electricity is stored in an energy storage device. When the wind is insufficient, that is, the electricity generated by the wind turbine is insufficient for the platform operation, the UPS provides 220 V power to the floating platform from the energy storage device. The energy storage device mainly consists of a system cabinet and a battery pack. The system cabinet (2400 mm × 800 mm × 1800 mm) contains rectifiers, inverters and bypasses. The battery pack (4153.9 mm × 600 mm × 1800 mm) is composed of a lithium iron phosphate battery, and its rated energy, rated voltage, and maximum discharge current are 188.16 kWh, 224 V, and 400 A, respectively. Power conversion from the wind turbine to the battery supply could be completed within 10 ms to ensure the continuous and normal operation of the floating platform.

**Current conversion module.** The intelligent current conversion module converts the AC power generated from the wind turbine into high-current DC power with low voltage. The maximum output voltage

is 8 V, and the maximum current is 1200 A. This approach provides a reliable and stable current for seawater electrolysis.

## Seawater electrolysis module

1.2 Nm³ H₂ h⁻¹ seawater electrolysis module, which consists of three seawater electrolysers, is the core component of the floating platform. The seawater cabin is connected to the ocean environment, and electrolysers are directly immersed in the seawater cabin for hydrogen generation. The electrolyser mainly consists of an outer frame, a "gas phase" isolation interface, and a core stack. The lower part of the outer frame (70.5 cm × 82 cm × 62 cm) is equipped with a porous grid. The inner wall of the outer frame is lined with a PTFE membrane, which forms a "gas phase" isolation interface in seawater and contains the electrolyte. The core electrolysis stack consists of 11 independent electrolysis units connected in parallel. Each unit includes a positioning frame, an anode plate, an anode catalyst layer, a hydrophilic diaphragm, and a cathode catalyst layer, and a cathode plate. A stainless-steel beam is welded to the outer frame of the electrolyser and fixed to the roof of the seawater chamber to resist swinging environment. A transparent PC board between the fixing beam of the electrolyser and steel beam in the seawater tank is used to seal the seawater cabin and support the electrolyser. Fluororubber gaskets are used at the connections for shock absorption, and they are connected to each other with bolts. Before testing, COMSOL was used to validate the design of the electrolyser.

**H₂ detection module.** The hydrogen generated from the electrolyser is transferred to the hydrogen detection module. The temperature of electronic condenser (a kind of gas-liquid separator) is set to 1.5 °C, and the setpoint accuracy is ±0.1 °C. After the hydrogen gas enters the electronic condenser, the alkaline steam from the hydrogen gas flow is condensed into a liquid solution, achieving gas-liquid separation as the temperature decreases. The condensed alkaline liquid is discharged through a drainage device, which works cyclically for 30 s and stops for 10 s after every 20 s. Approximately 700-900 ml of alkaline liquid was removed from a single electrolyser every 24 h. After cooling, the dehydrated hydrogen enters the detector for analysis.

## Strategies for overcoming crucial implementation challenges

Here, we provide an overview of the crucial challenges that were faced during the design, installation, and investigation of the kilowatt-scale floating seawater electrolysis platform and the mitigation strategies that were used to avoid or resolve each challenge. (a) The compatibility of the alkaline seawater electrolyser with fluctuating wind power. To obtain stable power from the wind turbine, a UPS with an energy storage system was installed. When the wind was sufficient, the UPS ensured a stable voltage output, while excess electricity was saved by the energy storage device, and when wind was insufficient, the energy storage device can provided stable power for the floating platform. (b) The connection and installation of the cable. To prevent the cables from sinking into the seabed and causing the floating platform to be dragged, a 30-m-long cable was secured to a series of floating piles. (c) Fixing the position of the floating platform. To prevent the floating platform from being dragged by cables or affected by strong winds and waves, a counterweight module and four anchors were installed on the floating platform body. (d) Corrosion protection. To prevent corrosion on the surface of the floating platform, a composite anti-corrosion coating was required on the floating system surface. In addition, it was necessary to install zinc blocks for cathodic protection on the platform. Moreover, considering salt mist and the relative humid (usually >80%) environment over the ocean, a dehumidification module had to be installed in the floating system to extend service life. (e) Hydrogen safety. Several vents and monitoring devices were installed inside the platform. In addition, a flowmeter was installed to monitor whether H₂ crossed over or leaked to stop electrolysis before dangerous conditions. (f) Alleviating the impact of waves. A seawater cabin directly linked to the external ocean via a control valve was constructed inside the floating platform. This design not only ensured direct interaction between the electrolyser and seawater but also enabled the floating structure (~48.5 t) to withstand the impact of waves. In addition, a sealed connection was required between the seawater cabin and electrolysers to prevent the seawater from sloshing and spilling into the interior of the floating platform. (g) Convenience and reliability. For ease of operation in an uncontrolled and fluctuating ocean environment, three lifting devices were installed in the floating system. This approach facilitated electrolyser installation, debugging, and emergency response procedures.

## Platform simulation and offshore testing in Xinghua Bay

To ensure the safety of the platform in the ocean, it was necessary to conduct a finite element simulation to analyse the stress in the platform in a fluctuating ocean environment. A model was established for the primary load-bearing components of the floating platform (the main body of the platform and four mooring racks). The simulation was carried out under the following conditions: a water depth of 10 m, a water density of 1025 kg m⁻³, force 8 wind (-17.2 m s⁻¹), an irregular wave with a peak period of 4 s, and an effective wave height of 1 m. The motion response and mooring force, as well as the stress distribution and fluid pressure of the platform under the action of wind and waves, were calculated to determine the kinematic stability and structural strength of the platform.

LSV was carried out to compare the electrolytic performance in the uncontrolled ocean environment with that in the laboratory. During the stability test, the voltage monitoring module collected the real-time voltage of the electrolyser with a high-conductivity and anticorrosive copper probe. The real-time voltage value was recorded every minute with an accuracy of ±0.01 V. A current sensing ring with an accuracy of ±1 A was used to monitor the real-time current of each electrolyser, and the current was recorded every minute. The hydrogen flow rate was detected via a gas monitoring device to determine whether the electrolyser was operating properly. All monitoring information was transmitted to the monitoring system cloud through 4G so that all the information could be obtained in real-time. In addition, SDE samples were collected daily from the electrolysers and tested for ion concentration to ensure that the water phase-transition migration process was not affected by wave fluctuations. The temperature, relative humidity, wave height, wind speed, and direction were recorded every 4 h.

## Gas analysis

H₂ produced by the electrolyser was dried and dealkalized. Ar gas was used as the carrier gas to push the collected H₂ into a gas chromatograph for the H₂ purity test. The contents of N₂, O₂, and CO₂ impurities were determined simultaneously, after which the purity of the hydrogen was calculated via the area normalisation method.

## Ion concentration test

The concentrations of $SO_4^{2-}$, $Cl^-$, $F^-$, and $Br^-$ ions were measured using an ion chromatograph (Thermo Scientific ICS100). $Mg^{2+}$, $Na^+$, $K^+$, $Ca^{2+}$, and $Sr^{2+}$ were analysed using an inductively coupled plasma optical emission spectrometer (PE Avio 200 ICP–OES). The hypochlorite ($ClO^-$) concentration in the SDE was measured by the o-tolidine method[31].

## Electrolytic energy consumption associated with H₂ production

The energy consumption for producing hydrogen per Nm³ of H₂ of the three electrolysers, $Q$, was calculated as follows:

$$W = I \times \int U dt$$

where $I$ is the current and $U$ is the voltage. $W$ of the three electrolysers corresponds to $W_1$, $W_2$ and $W_3$. Next, the volume of $H_2$ produced, $V$, was calculated as follows:

$$V = 22.4 \times I \times t / (Z \times F)$$

where the number of electrons transferred for the hydrogen evolution reaction, $Z$, is 2 and the Faraday constant, $F$, is 96,485 C mol$^{-1}$. $V$ of the three electrolysers correspond to $V_1$, $V_2$ and $V_3$.

Thus, the electrolytic energy consumption is $Q = W/V$ (kWh Nm$^{-3}$), and the average electrolytic energy consumption is $Q = (W_1 + W_2 + W_3)/(V_1 + V_2 + V_3)$.

## Model

The theoretical model was based on the following assumptions: (1) In the instantaneous process, temperature changes, thermal convection, and concentration polarization on both sides of the membrane are not considered. (2) The amount of water migration per unit time is equal to that absorbed by the KOH solution (SDE) for the same area and time.

According to Darcy's law, the water migration mass flux is assumed to be proportional to the vapour pressure difference across the PTFE membrane and is given by (1):

$$J(x) = B_m \Delta P = B_m \left[ P_S - P(x) \right] \tag{1}$$

where $J(x)$ is the mass flux (g m$^{-2}$ s$^{-1}$), $B_m$ is the membrane permeability (g m$^{-2}$ s$^{-1}$ Pa$^{-1}$), and $P_s$ and $P(x)$ are the water vapour pressures at the seawater and the SDE side surfaces (Pa), respectively.

The water migration rate and water migration mass can be calculated with (2) and (3):

$$\left( \frac{dM}{dt} \right)_{migration} = J(x)S = B_m \left[ P_s - P(x) \right] S \tag{2}$$

$$M_{migration} = J(x) \int_0^t S = B_m \left[ P_s - P(x) \right] \int_0^t S \tag{3}$$

Seawater can be considered as a NaCl solution. The water vapour pressure on the seawater side can be calculated with (4)[46]:

$$P_s = (1 - x_s) \times (1 - 0.5 x_s - 10 x_s^2) \times \exp \left[ 23.1964 - \frac{3816.44}{T - 46.13} \right] \tag{4}$$

where $x_s$ is the molar fraction of NaCl.

The water vapour pressure on the SDE side can be calculated with (5)[47]:

$$
\begin{aligned}
\lg \left( \frac{P(x)}{10^5} \right) = & \left[ -0.01508 \times \left( \frac{55.56x}{1-x} \right) - 0.0016788 \times \left( \frac{55.56x}{1-x} \right)^2 \right. \\
& \left. + 2.25887 \times 10^{-5} \times \left( \frac{55.56x}{1-x} \right)^3 \right] \\
& + \left[ 1 - 0.0012062 \times \left( \frac{55.56x}{1-x} \right) + 5.6024 \times 10^{-4} \right. \\
& \times \left( \frac{55.56x}{1-x} \right)^2 - 7.8228 \times 10^{-6} \left( \frac{55.56x}{1-x} \right)^3 \right] \\
& \times \left( 35.4462 - \frac{3343.93}{T} - 10.9 \lg T + 0.0041645T \right)
\end{aligned}
\tag{5}
$$

where, $x$ is the molar fraction of KOH solution.

The water consumption during electrolysis can be calculated with Eqs. (6) and (7):

$$It = nZF \tag{6}$$

$$M_{electrolysis} = \frac{18It}{zF} \tag{7}$$

where $I$ is the current (A), $z$ is the number of electrons transferred for the HER, and $F$ is the Faraday constant (96,485 C mol$^{-1}$).

When the water consumption for water migration and electrolysis are equal, the system becomes balanced.

$$\Delta M = M_{migration} - M_{electrolysis} = B_m \left[ P_s - Px \right] \int_0^t S - \frac{18It}{zF} \tag{8}$$

## Reporting summary

Further information on research design is available in the Nature Portfolio Reporting Summary linked to this article.

## Data availability

The data supporting the findings of this study are available with the article and its supplementary files. Any additional requests for information can be directed to, and will be fulfilled by, the corresponding authors. Source data are provided in this paper. Source data are provided with this paper.

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

## Acknowledgements

This work is supported by Program for Guangdong Introducing Innovative and Entrepreneurial Teams (Grant No. 2019ZT08G315 to H.X.). We would like to thank the National Natural Science Foundation of China (Grant No. 52104400 to Y.W., No. 52004166 to T.L., No. 52304427 to C.L., No. 51827901 to H.X., No. 52374133 to Y.W.). We are grateful for the support from the National Key R&D Program of China (Grant No. 2022YFB4102100 to T.L.). We also thank the Institute of New Energy and Low-Carbon Technology (Sichuan University), Dongfang Electric Wind Power Co., Ltd, China Three Gorges Corporation Fujian Branch for support,

## Author contributions

H.X., T.L., T.S.L., Z.Z., and Y.C. conceived and designed the floating electrolysis system. T.L., Z.Z., W.T., Y.C. C.L., L.Z., W.J., Y.W., Y.W., Z.Y., and D.Y. performed the experiments on the Xinghua Bay, Fujian province. T.L., Z.Z., and W.T. performed characterizations and laboratory experiments. T.L. and Z.Z. performed the modelling. H.X. and T.S.L. managed and supervised the project. Q.W. and L.L. provided the wind turbine and marine experimental conditions. Z.Z., T.L. W.T., and H.X. wrote the manuscript. All authors reviewed the manuscript.

## Competing interests

The authors declare no competing interests.
