## [Peer Review File · Nature Communications]

In-situ direct seawater electrolysis using floating platform in ocean with uncontrollable wave motionREVIEWER COMMENTS

Reviewer #1 (Remarks to the Author):

1. What are the noteworthy results?

In this study, expanding upon the design of the lab-scale demonstration model, a large-scale and efficient floating system of in-situ direct seawater electrolysis, based on the water phase-transition migration mechanism under interfacial vapour pressure difference, using renewable offshore wind energy, was operated in the ocean with uncontrollable fluctuating wind and wave conditions for the first time.

2. Will the work be of significance to the field and related fields? How does it compare to the established literature?

Yes, this work will be of significance to the field due to the transition from lab-scale demonstration models to large-scale functional ones in real conditions of waves and wind.

Hydrogen production directly from natural resources like seawater using renewable energies offers a promising pathway to achieve a sustainable energy industry and fuel economy as evidenced by the large number of articles in this area of research. Various direct seawater electrolysis methods have demonstrated effective at the lab-scale.

3. Does the work support the conclusions and claims, or is additional evidence needed?

There is no additional evidence needed

4. Are there any flaws in the data analysis, interpretation and conclusions? Do these prohibit publication or require revision?

After the Abstract from paragraphs 21-38, it would be indicated to appear the "Introduction" section for paragraphs 42-104.

In my opinion, the conclusions should be outlined in a separate "Conclusions" section.

5. Is the methodology sound? Does the work meet the expected standards in your field?

Throughout the manuscript there are interpretations or discussions repeated several times, which makes reading difficult.

Also, the methods and analyses used in the laboratory-scale model are not presented separately from those applied to the real model.

6. Is there enough detail provided in the methods for the work to be reproduced?

In my opinion, it is a problem that many details and information are provided in the Supplementary file, so that there are always references in the text to this additional material. Perhaps it would be advisable for the additional material to contain only figures, and the explanations to be given in the article, i.e. the complete model and pages 11-12.

Reviewer #2 (Remarks to the Author):

In-situ direct seawater electrolysis in uncontrollable fluctuating ocean

This article studies the electrolysis of seawater without purification in a floating platform subject to the motion caused by the waves, the wind and currents. The topic is interesting for the audience of Nature Communications. However, it requires substantial improvement prior to being accepted for publication.

Regarding the title, it should be a composition of keywords relevant to the article. The ocean is uncontrollable and fluctuating, so the two descriptive adjectives that appear in the title add no information. I would suggest including the words "floating platform" and "wave motion" in the title. About the arrangement of the article. Right now, the article presents the following structure:

- Abstract
- Results
 - o Process variable correlations and model
 - o System overview
 - o Performances in uncontrollable fluctuating ocean

- o Potential and outlook
- Discussion
- Methods
- o Pilot plant design and operation
- ♣ UPS module
- ♣ Current conversion module
- ♣ Seawater electrolysis module
- ♣ H2 detection module
- o Integrated testing
- o Electrochemical test
- o Ion concentration test
- o Material characterization
- o Gas analysis
- o Model
- Reference
- Acknowledgements
- Author contributions
- Competing interests

The current structure must be improved. It is not possible for the reader to jump directly into the results. The article must have first a comprehensive literature review in the Introduction to state the problem this article is going to solve and why no other previous research has been able to tackle the specific problem. The Introduction shall also include the scope of the article. The introduction must include as well a summary of the structure of the article at the end.

Furthermore, the introduction should include what the reader should expect to find in the main body of the article and what could find in the supplementary information document. Normally after the introduction comes the Methods, then the results, the discussion and the conclusions.

Currently there is no conclusions section.

The supplementary information document arrangement must include an index in addition to the index of figures. The supplementary information document must include an introduction to explain the reader the structure of the document. The article shall be able to stand alone without the need to refer to the supplementary information document. Right now it is not possible to read without consulting the supplementary information document. The supplementary information document must be able to stand alone as well. The arrangement of the main body of the article and the supplementary information document must be revised.

Regarding the physics modelling of the floating platform, wave characteristics are not just defined by amplitude. In addition, the system is dynamic, so it is important to consider other parameters like the moments of inertia, the masses, the added masses and the hydrostatics of the platform. I suggest the authors to carefully read the following references:

- Edition, F., Journée, J. M. J., & Massie, W. W. (2001). Offshore hydromechanics. Delft University of Technology.
- Chakrabarti, S. (2005). Handbook of Offshore Engineering (2-volume set). Elsevier.

I suggest providing the orientation of the floating platform and the directional rose of waves and winds to complete all the information.

About the references to previous works. The article presents only 23 references, out of them none include DOI. Two of them correspond to press releases, in which the URL of the article is provided but no the accession date. The literature review must be much more extensive to state the problem and the rationale to this work.

Concerning the reproducibility of the research. In the current status the research described in the article is impossible to reproduce, as the current information is not properly organized and it is incomplete. The authors must improve the article to ensure that the research could be reproduced by any other researcher. In addition, some of the problem that this article covers are not properly described. It is not possible for the reader to understand problems like the water migration problem in Figure 1. The electrolysis cell is not properly described, thus, it is not possible to understand their architecture, and how seawater is separated from the SDE. In Figure 2, what is the gas condenser? Is it actually a moisture remover?.

About the units in the results. You can provide results in any units you wish but SI units are mandatory. Use MJ for kWh and use kg for Nm³.

About the English language: Proof-reading is highly recommended.

Happy New Year!

Reviewer #3 (Remarks to the Author):

Manuscript number: NCOMMS-23-55237

The manuscript entitled " In-situ direct seawater electrolysis in uncontrollable fluctuating ocean" addressed an important and interesting research study. It is significant to explore to potential opportunities to employ direct seawater electrolysis in the offshore environment. Fluctuating conditions of the ocean are considered in the model development to guide the in-situ direct electrolysis for hydrogen generation in the fluctuating ocean. The methodology is sound and results are noteworthy and support the conclusions drawn from the manuscript, however, further clarification, justification and additional results would help consolidate the idea being presented. Some critical points as follows need to be addressed:

Additional details need to be added on sizing of the proposed model. On what basis $1.2 \text{ Nm}^3\text{h}^{-1}$ scale was selected?

A 240-h stable electrolysis operation with an energy consumption of $5 \text{ kWh Nm}^{-3} \text{ H}_2$. How is this energy consumption is determined and how the components such as seawater electrolysis and energy storage behave in real-time on the fluctuation in wind power?

Authors assumed the wind speed range of 0-15 m/s. How will the model behave if wind speed increases from the assumed limit?

Some insights and atleast two cases would be required how the modeled system is producing $1.2 \text{ Nm}^3\text{h}^{-1}$ of hydrogen when the input power source is intermittent in nature, how imperative part is being played by the energy storage system and some details on the storage system sizing would add value.

SEM images are shows for the catalysts and PTFE membrane after 10 days seawater electrolysis, however, it would be imperative to see the long term effects on the electrodes, catalysts and membrane to predict the lifetime.

Point-by-point responses to the reviewers' comments

Referee #1

Comments:

1. What are the noteworthy results?

In this study, expanding upon the design of the lab-scale demonstration model, a large-scale and efficient floating system of in-situ direct seawater electrolysis, based on the water phase-transition migration mechanism under interfacial vapour pressure difference, using renewable offshore wind energy, was operated in the ocean with uncontrollable fluctuating wind and wave conditions for the first time.

Response:

We would like to thank the reviewer very much for his/her positive comments and precise summary of the noteworthy results of our work.

Comments:

2. Will the work be of significance to the field and related fields? How does it compare to the established literature?

Yes, this work will be of significance to the field due to the transition from lab-scale demonstration models to large-scale functional ones in real conditions of waves and wind.

Hydrogen production directly from natural resources like seawater using renewable energies offers a promising pathway to achieve a sustainable energy industry and fuel economy as evidenced by the large number of articles in this area of research. Various direct seawater electrolysis methods have demonstrated effective at the lab-scale.

Response:

We are very grateful to the reviewer for his/her positive comments on our work of direct seawater electrolysis in fluctuating ocean, which is greatly encouraging us. This work is a new exploration on the field of seawater electrolysis from the laboratory to real ocean environments, which could provide a solution for developing a more mature mode of hydrogen production directly from seawater using renewable energy. This approach is expected to promote the efficient application of offshore renewable energy and reduce the cost of large-scale hydrogen fuel production in the future. In addition, this work may guide researchers to pay more attention to the impact of the ocean environment on seawater electrolysis, thereby promoting the development of this field towards to industrial application.

Comments:

3. Does the work support the conclusions and claims, or is additional evidence needed?

There is no additional evidence needed.

Response:

We would like to thank the reviewer for his/her recognition of the adequacy and comprehensiveness of our work.

Comments:

4. Are there any flaws in the data analysis, interpretation and conclusions? Do these prohibit publication or require revision?

After the Abstract from paragraphs 21-38, it would be indicated to appear the "Introduction" section for paragraphs 42-104.

In my opinion, the conclusions should be outlined in a separate "Conclusions" section.

Response:

We apologize that the title "Introduction" was not included in the original manuscript. In the revised manuscript, the heading "Introduction" was added to differentiate for paragraphs (**in line 42-106**) after the "Abstract" section. Additionally, the "Conclusion" of our manuscript is included in the "Discussion" section (**in line 325-353**) by following the style of the most recent literature appearing in *Nature Communications* and "Formatting Instructions" given by *Nature Communications* (Liu, Z. et al. Synergistic dual-phase air electrode enables high and durable performance of reversible proton ceramic electrochemical cells. *Nat Commun* **15**, 472 (2024); Sun, M. et al. Gas diffusion enhanced electrode with ultrathin superhydrophobic macropore structure for acidic CO₂ electroreduction. *Nat Commun* **15**, 491 (2024); Zhang, X. et al. A dicarbonate solvent electrolyte for high performance 5 V-Class Lithium-based batteries. *Nat Commun* **15**, 536 (2024); Meng, J. et al. Rapid-charging aluminium-sulfur batteries operated at 85 °C with a quaternary molten salt electrolyte. *Nat Commun* **15**, 596 (2024)).

Comments:

5. Is the methodology sound? Does the work meet the expected standards in your field?

Throughout the manuscript there are interpretations or discussions repeated several times, which makes reading difficult.

Also, the methods and analyses used in the laboratory-scale model are not presented separately from those applied to the real model.

Response:

We apologize that the repeated interpretations or discussions in our original manuscript made reading more difficult. In the revised manuscript, we have revised the content carefully to make the manuscript more concise and easier to read. In addition, we have added “Seawater electrolysis in a simulated fluctuating environment in the laboratory” to the “Methods” section to separate the laboratory-scale electrolysis from the real model in ocean and added more details to make our work more solid and comprehensive (in line 371-384).

‘To test the durability of the seawater electrolyser in a fluctuating seawater environment, an open $1.4 \times 1.2 \times 0.6 \text{ m}^3$ container containing seawater from Xinghua Bay was used as the seawater cabin, and irregular waves were created to simulate the fluctuating environment of the ocean. The seawater electrolyser was immersed in seawater with a specific water migration area according to the concentration of seawater and SDE. Linear sweep voltammetry (LSV) was conducted using a multimeter. The corresponding voltages between the anode and cathode plates of the electrolyser were obtained at current densities of 0 mA cm^{-2} , 10 mA cm^{-2} , 50 mA cm^{-2} , 100 mA cm^{-2} , 150 mA cm^{-2} , 200 mA cm^{-2} , and 259 mA cm^{-2} . The electrolysers were operated for more than 500 h at a current of 957 A. The voltage of each electrolyser was recorded as the result of the electrolysis process. After electrolysis, the microstructure of the gas pathways in the PTFE membrane and the morphology of the commercial catalyst were observed via SEM analysis (Thermo Scientific Apreo 2C). The water contact angle of the PTFE membrane was measured at room temperature using a contact angle goniometer (DSA 25, Krüss) seawater droplets.’

Comments:

6. Is there enough detail provided in the methods for the work to be reproduced?

In my opinion, it is a problem that many details and information are provided in the Supplementary file, so that there are always references in the text to this additional material. Perhaps it would be advisable for the additional material to contain only figures, and the explanations to be given in the article, i.e. the complete model and pages 11-12.

Response:

We would like to thank the reviewer for his/her comment on our supplementary materials. In response to the reviewer’s suggestion, some important explanations, such as the complete model and “Strategies for overcoming crucial implementation challenges”, have been added to the “Methods” section of the revised manuscript rather than in the supplementary information. We have condensed and revised the relevant content for improved readability. In addition, some figures originally in supplementary information also have been placed in the revised manuscript (Fig. R1_1 and Fig. R1_2), so that the manuscript contains more useful information.

Fig. R1_1 | Overview of the platform. **a**, Illustration of the overall system consisting of the UPS module, current conversion module, seawater electrolysis module, H₂ detection module and transportation module. **b**, Process diagram of the device (I =current value, U =voltage value, Q =flow value). **c**, Photo of the interior space of the floating platform. **d**, Diagram of seawater cabin connected with ocean.

Fig. R1_2 | Durability test in uncontrollable fluctuating ocean. **a**, Variation in the temperature and humidity in Xinghua Bay. The data were recorded every 4 hours. The wind speed (**b**) and wave height (**c**) during the period of May 17, 2023 to May 26, 2023. The data were recorded every 4 hours. **d**, Stability of the seawater electrolysis system in uncontrollable fluctuating ocean. **e**, The ions concentration of SDE and Xinghua Bay seawater. SEM images of catalyst (**f**) and PTFE membrane (**g**) after electrolysis in fluctuating ocean. **h**, Contact angle of PTFE membrane after electrolysis. **i**, Water migration behaviour using the PTFE membrane before and after 10 days used. All error bars indicate the standard deviation at three measurements.

Referee #2

Comments:

1. In-situ direct seawater electrolysis in uncontrollable fluctuating ocean

This article studies the electrolysis of seawater without purification in a floating platform subject to the motion caused by the waves, the wind and currents. The topic is interesting for the audience of Nature Communications. However, it requires substantial improvement prior to being accepted for publication.

Response:

We are very grateful to the reviewer for his/her positive evaluation of our work. We have extensively revised the structure and content of our manuscript based on his/her comments to enhance the research content and viewpoints presented in our articles.

Comments:

2. Regarding the title, it should be a composition of keywords relevant to the article. The ocean is uncontrollable and fluctuating, so the two descriptive adjectives that appear in the title add no information. I would suggest including the words “floating platform” and “wave motion” in the title:

Response:

We would like to thank the reviewer for his/her advice about the title. We have seriously considered the reviewer's suggestion and highlighted key words in the title. The new title is as follows:

'In-situ direct seawater electrolysis using floating platform in ocean with uncontrollable wave motion'

Comments:

3. About the arrangement of the article. Right now, the article presents the following structure:

- Abstract
- Results
 - o Process variable correlations and model
 - o System overview
 - o Performances in uncontrollable fluctuating ocean
 - o Potential and outlook
- Discussion
- Methods
 - o Pilot plant design and operation
 - UPS module
 - Current conversion module
 - Seawater electrolysis module
 - H₂ detection module

- o Integrated testing
- o Electrochemical test
- o Ion concentration test
- o Material characterization
- o Gas analysis
- o Model
- Reference
- Acknowledgements
- Author contributions
- Competing interests

The current structure must be improved. It is not possible for the reader to jump directly into the results. The article must have first a comprehensive literature review in the Introduction to state the problem this article is going to solve and why no other previous research has been able to tackle the specific problem. The Introduction shall also include the scope of the article. The introduction must include as well a summary of the structure of the article at the end. Furthermore, the introduction should include what the reader should expect to find in the main body of the article and what could find in the supplementary information document. Normally after the introduction comes the Methods, then the results, the discussion and the conclusions. Currently there is no conclusions section.

Response:

We would like to thank the reviewer for his/her helpful comment, which is important in improving the readability of our work.

In response to reviewer's suggestion, the article structure has been revised and the "Conclusion" section of our manuscript is included in the "Discussion" section by following the style of the most recent literature appearing in *Nature Communications* and "Formatting Instructions" given by *Nature Communications* (Liu, Z. et al. Synergistic dual-phase air electrode enables high and durable performance of reversible proton ceramic electrochemical cells. *Nat Commun* **15**, 472 (2024); Sun, M. et al. Gas diffusion enhanced electrode with ultrathin superhydrophobic macropore structure for acidic CO₂ electroreduction. *Nat Commun* **15**, 491 (2024); Zhang, X. et al. A dicarbonate solvent electrolyte for high performance 5 V-Class Lithium-based batteries. *Nat Commun* **15**, 536 (2024); Meng, J. et al. Rapid-charging aluminium-sulfur batteries operated at 85 °C with a quaternary molten salt electrolyte. *Nat Commun* **15**, 596 (2024)).

The new structure of our manuscript is as follows:

• **Abstract**

• **Introduction**

• **Results**

o *Process variable correlations during seawater electrolysis*

o *Floating platform system overview*

o *Performance under uncontrolled fluctuating ocean conditions*

o *Potential and outlook*

• **Discussion**

• **Methods**

o *Water migration test in the laboratory*

- o Seawater electrolysis in a simulated fluctuating environment in the laboratory*
- o Floating platform design*
- o Strategies for overcoming crucial implementation challenges*
- o Platform simulation and offshore testing in Xinghua Bay*
- o Gas analysis*
- o Ion concentration test*
- o Electrolytic energy consumption associated with H₂ production*
- o Model*
- **Reference**
- **Acknowledgements**
- **Author contributions**
- **Competing interests'**

In addition, we apologize that the title of the “Introduction” was not previously included in the original manuscript, which led to a misunderstanding by the reviewers. We have revised and improved the content of the introduction according to reviewer’s advice, including the significance of our work, overview and issues in this field, innovation of the article, and summary of the article (**in line 42-106**).

Comments:

4. The supplementary information document arrangement must include an index in addition to the index of figures. The supplementary information document must include an introduction to explain the reader the structure of the document. The article shall be able to stand alone without the need to refer to the supplementary information document. Right now it is not possible to read without consulting the supplementary information document. The supplementary information document must be able to stand alone as well. The arrangement of the main body of the article and the supplementary information document must be revised.

Response:

We would like to thank the reviewer for his/her suggestions on our supplementary information section. We have added the structure explanation to the first page of supplementary information file (**in line 24-29**), including the title, author information, table of contents, supplementary figures and references to illustrate the content of each section. This information was based on the most recent paper appearing in Nature Communications (Zhong, T. et al. A wireless battery-free eye modulation patch for high myopia therapy. *Nat Commun* **15**, 1766 (2024); Peng, Q. et al. Extreme Li-Mg selectivity via precise ion size differentiation of polyamide membrane. *Nat Commun* **15**, 2505 (2024)). Furthermore, a catalogue was created to clearly list the figures and their page numbers.

In addition, we apologize that the original supplementary information contains much information. To make our manuscript clearer, some necessary descriptions and data that were originally included in the supplementary information, including the complete model, the strategies for overcoming challenges, and important data, have been added to the revised manuscript, and relevant explanations have been provided. In addition, related data in the supplementary information have been combined to make the file more readable (as shown in **Fig. R2_1** and **Fig. R2_2**).

Fig. R2_1 | Overview of the platform. **a**, Illustration of the overall system consisting of the UPS module, current conversion module, seawater electrolysis module, H₂ detection module and transportation module. **b**, Process diagram of the device (I =current value, U =voltage value, Q =flow value). **c**, Photograph of the interior space of the floating platform. **d**, Diagram of the seawater cabin connected to the ocean.

Fig. R2_2 | Durability test in an uncontrollable fluctuating ocean environment. **a**, Variation in the temperature and relative humidity in Xinghua Bay. The data were recorded every 4 hours. The wind speed (**b**) and wave height (**c**) during the period from May 17, 2023 to May 26, 2023. The data were recorded every 4 hours. **d**, Stability of the seawater electrolysis system in an uncontrolled fluctuating ocean environment. **e**, The ion concentrations of SDE and Xinghua Bay seawater. SEM images of the catalyst (**f**) and PTFE membrane (**g**) after electrolysis in a fluctuating ocean. **h**, Contact angle of the PTFE membrane after electrolysis. **i**, Water migration behaviour of the PTFE membrane before and after 10 days of use. All error bars indicate the standard deviation at three measurements.

Comments:

5. Regarding the physics modelling of the floating platform, wave characteristics are not just defined by amplitude. In addition, the system is dynamic, so it is important to consider other parameters like the moments of inertia, the masses, the added masses and the hydrostatics of the platform. I suggest the authors to carefully read the following references:

- Edition, F., Journée, J. M. J., & Massie, W. W. (2001). Offshore hydromechanics. Delft University of Technology.
- Chakrabarti, S. (2005). Handbook of Offshore Engineering (2-volume set). Elsevier.

Response:

We are very grateful to the reviewer for his/her suggestions on the physical modelling of our floating platform, which is important for improving our work.

For the **laboratory-scale model**, the characteristic parameters of the simulated waves, including the wave period and wavelength, were considered in the water migration behaviour test based on the reviewer's suggestion. According to the experiment, the period of different simulated waves is approximately 1.34 s, while the wave heights of 3.4, 3.2 and 2.9 cm correspond to the wavelengths of 62.32, 59.40 and 54.52 cm, respectively. We have added a relevant description of the parameters of the simulated wave in the manuscript (**in line 129-132**):

'The average migration rate of water at the different wave heights and lengths significantly increased. The water migration rates were approximately 0.0136, 0.0177 and 0.0194 g_{H2O} g_{SDE}⁻¹ min⁻¹ at wave heights of 2.9, 3.2 and 3.4 cm, as well as the corresponding wave lengths of 54.5, 59.4 and 62.3 cm, respectively'

For **physical modelling of the floating platform**, the motion and surface pressure distribution of the floating platform under the comprehensive action of wind and waves were simulated by finite element analysis. The geometric model of the platform was composed of a platform body (7 m × 9 m) and four mooring racks (**Fig. R2_3a**), in which the depth was 2.5 m and the draft depth was 1 m. The floating platform was symmetrical about the xz and yz planes, and its moment of inertia I_{xx}, I_{yy} and I_{zz} about the coordinate axes are 76798, 411172 and 592647 kg m², respectively. The mooring arrangement of the platform was mainly composed of four moored cables at four corners (**Fig. R2_3b**). The simulation was carried out under the following conditions: a water depth of 10 m; a water density of 1025 kg m⁻³; a force 8 wind (~17.2 m s⁻¹); and an irregular wave with a peak period of 4 s and an effective significant wave height of 1 m.

Recording the evolution of the six degrees of freedom of the platform enabled determination of the platform's motion transformation over time (**Fig. R2_4**). Under the action of wind and waves, the floating platform remained stable between 5.5 and 7.9 m in the X direction (sway motion) under the action of the mooring cable. The displacement of the platform in the Y direction (surge motion) was within the range of -0.55 to 0.56 m, and the displacement in the Z direction (heave motion) was within the range of -0.54 to 0.9 m. The floating platform deviated from -2.5 to 2.0° around the X-axis (roll), from -8.5 to 7.2° around the Y-axis (pitch), and from -12.4 to 10.2° around the Z-axis (yaw). To analyse the structural stability of the platform, we calculated the stress distribution and fluid pressure of the platform body when the floating platform was subjected to a maximum mooring force of 73.2 kN. The maximum stress of the platform was located at mooring rack with 286 MPa, and the maximum fluid pressure on the wet surface was 13041 Pa (**Fig. R2_5**). The forces are less than the yield stress of the floating structure steel, which could ensure the stability of platform.

We have added the corresponding description of the platform analysis in the manuscript as follows (**in line 236-246**):

'To ensure the stability and safety of the floating platform in the ocean, the finite element simulation was used to analyse the motion, pressure and stress distribution of the floating platform under a force 8 wind (~ 17.2 m s⁻¹) and a 1-m significant wave height. The floating platform, stabilized by mooring ropes, could swing between 5.5 and 7.9 m along the X direction (sway motion), while it could surge

and heave along the Y direction (-0.55~0.56 m) and Z direction (0.54~0.9 m), respectively. The floating platform rotated from -2.5 to 2.0° around the X-axis (roll), from -8.5 to 7.2° around the Y-axis (pitch), and from -12.4 to 10.2° around the Z-axis (yaw), which was within a controllable range (**Supplementary Fig. 9**). In addition, as shown in **Fig. 3c**, maximum stress occurs at the anchor frame (~286 MPa), and the maximum fluid pressure on the wet surface of the floating platform body was approximately 13041 Pa. These values were lower than the yield stress of the ship structural steel employed, ensuring adequate strength for the floating platform.'

Fig. R2_3. a, Floating platform model. b, Simulation conditions.

Fig. R2_4. Motion distance in the X direction (sway) (a), Y direction (surge) (b) and Z direction (heave) (c). Deviation angle in the X direction (roll) (d), Y direction (pitch) (e) and Z direction (yaw) (f).

Fig. R2_5. a, The stress distribution of the floating platform. **b,** Hydrodynamic pressure distribution on the wet surface of the floating platform.

Comments:

6. I suggest providing the orientation of the floating platform and the directional rose of waves and winds to complete all the information.

Response:

We are very grateful to the reviewer for his/her suggestions on the details of our study.

The orientation figure of the floating platform is shown in **Fig. R2_6a** (also shown in **Fig. 3d**). The floating platform is about southeast of the wind turbine, and it is connected by cables. We have added the relevant description in manuscript as follows (in line 247):

‘The floating platform is about southeast of the wind turbine’

The rose diagram of wind directional based on wind direction data recorded every 4 h is shown in **Fig. R2_6b** (also shown in **Fig. 3e**). And we have added the relevant description in manuscript as follows (in line 259-260):

‘During this period, winds are primarily from the northeast and southwest’

Fig. R2_6. a, Orientation diagram of floating platform in ocean. **b,** Directional rose chart of wind during offshore test.

Comments:

7. About the references to previous works. The article presents only 23 references, out of them none include DOI. Two of them correspond to press releases, in which the URL of the article is provided but no the accession date. The literature review must be much more extensive to state the problem and the rationale to this work.

Response:

We apologize that the references were not clear or accurate. According to the “Formatting Instructions” given by *Nature* and the reviewer’s suggestion, we have added more previous work about direct seawater electrolysis, and revised the format of all references (in line 537-674). In addition, more references about seawater electrolysis have been added to the manuscript:

20. Fan R, Liu C, Li Z, et al. Ultrastable electrocatalytic seawater splitting at ampere-level current density. *Nature Sustainability* **7**, 158-167 (2024). <https://doi.org/10.1038/s41893-023-01263-w>
21. Liu X, Yu Q, Qu X, et al. Manipulating electron redistribution in Ni₂P for enhanced alkaline seawater electrolysis. *Advanced Materials* **36**, 2307395 (2024). <https://doi.org/10.1002/adma.202307395>
22. Li P, Zhao S, Huang Y, et al. Corrosion resistant multilayered electrode comprising Ni₃N nanoarray overcoated with NiFe-Phytate complex for boosted oxygen evolution in seawater electrolysis. *Advanced Energy Materials* **14**, 2303360 (2024). <https://doi.org/10.1002/aenm.202303360>
23. Xu W, Wang Z, Liu P, et al. Ag nanoparticle-induced surface chloride immobilization strategy enables stable seawater electrolysis. *Advanced Materials* **36**, 2306062 (2024). <https://doi.org/10.1002/adma.202306062>
24. Zhou L, Guo D, Wu L, et al. A restricted dynamic surface self-reconstruction toward high-performance of direct seawater oxidation. *Nature Communications* **15**, 2481 (2024). <https://doi.org/10.1038/s41467-024-46708-8>
25. Duan X, Sha Q, Li P, et al. Dynamic chloride ion adsorption on single iridium atom boosts seawater oxidation catalysis. *Nature Communications* **15**, 1973 (2024). <https://doi.org/10.1038/s41467-024-46140-y>
26. You H, Wu D, Si D, et al. Monolayer NiIr-layered double hydroxide as a long-lived efficient oxygen evolution catalyst for seawater splitting. *Journal of the American Chemical Society* **144**, 9254-9263 (2022). <https://doi.org/10.1021/jacs.2c00242>
27. Tan L, Yu J, Wang C, et al. Partial sulfidation strategy to NiFe-LDH@ FeNi₂S₄ heterostructure enable high-performance water/seawater oxidation. *Advanced Functional Materials* **32**, 2200951 (2022). <https://doi.org/10.1002/adfm.202200951>
28. Yu L, Zhu Q, Song S, et al. Non-noble metal-nitride based electrocatalysts for high-performance alkaline seawater electrolysis. *Nature Communications* **10**, 5106 (2019). <https://doi.org/10.1038/s41467-019-13092-7>
29. Keane T P, Veroneau S S, Hartnett A C, et al. Generation of pure oxygen from briny water by binary catalysis. *Journal of the American Chemical Society* **145**, 4989-4993 (2023). <https://doi.org/10.1021/jacs.3c00176>
30. Dresp S, Thanh T N, Klingenhof M, et al. Efficient direct seawater electrolyzers using selective alkaline NiFe-LDH as OER catalyst in asymmetric electrolyte feeds. *Energy & Environmental Science* **13**, 1725-1729 (2020). <https://doi.org/10.1039/d0ee01125h>

31. Sun F, Qin J, Wang Z, et al. Energy-saving hydrogen production by chlorine-free hybrid seawater splitting coupling hydrazine degradation. *Nature Communications* **12**, 4182 (2021). <https://doi.org/10.1038/s41467-021-24529-3>
32. Zhang L, Wang Z, Qiu J. Energy-saving hydrogen production by seawater electrolysis coupling sulfion degradation. *Advanced Materials* **34**, 2109321 (2022). <https://doi.org/10.1002/adma.202109321>
33. Xin Y, Shen K, Guo T, et al. Coupling hydrazine oxidation with seawater electrolysis for energy-saving hydrogen production over bifunctional CoNC nanoarray electrocatalysts. *Small* **19**, 2300019 (2023).
34. Guo L, Chi J, Zhu J, et al. Dual-doping NiMoO₄ with multi-channel structure enable urea-assisted energy-saving H₂ production at large current density in alkaline seawater. *Applied Catalysis B: Environmental* **320**, 121977 (2023). <https://doi.org/10.1016/j.apcatb.2022.121977>
35. Zhu D, Zhang H, Miao J, et al. Strategies for designing more efficient electrocatalysts towards the urea oxidation reaction. *Journal of Materials Chemistry A* **10**, 3296-3313 (2022). <https://doi.org/10.1039/D1TA09989B>
36. Jiang X, Dong Z, Zhang Q, et al. Decoupled hydrogen evolution from water/seawater splitting by integrating ethylene glycol oxidation on PtRh_{0.02}@Rh nanowires with Rh atom modification. *Journal of Materials Chemistry A* **10**, 20571-20579 (2022). <https://doi.org/10.1039/D2TA05469H>
37. Zhu W, Wei Z, Ma Y, et al. Energy-efficient electrosynthesis of high value-added active chlorine coupled with H₂ generation from direct seawater electrolysis through decoupling electrolytes. *Angewandte Chemie*, e202319798 (2024). <https://doi.org/10.1002/ange.202319798>
38. Liu K, Gao X, Liu C X, et al. Energy-saving hydrogen production by seawater splitting coupled with PET plastic upcycling. *Advanced Energy Materials*, 2304065 (2024). <https://doi.org/10.1002/aenm.202304065>
39. Shi L, Rossi R, Son M, et al. Using reverse osmosis membranes to control ion transport during water electrolysis. *Energy & Environmental Science* **13**, 3138-3148 (2020). <https://doi.org/10.1039/D0EE02173C>
40. Veroneau S S, Nocera D G. Continuous electrochemical water splitting from natural water sources via forward osmosis. *Proceedings of the National Academy of Sciences* **118**, e2024855118 (2021). <https://doi.org/10.1073/pnas.2024855118>
41. Veroneau S S, Hartnett A C, Thorarinsdottir A E, et al. Direct seawater splitting by forward osmosis coupled to water electrolysis. *ACS Applied Energy Materials* **5**, 1403-1408 (2022). <https://doi.org/10.1021/acsaem.1c03998>
42. Marin D H, Perryman J T, Hubert M K A, et al. Hydrogen production with seawater-resilient bipolar membrane electrolyzers. *Joule* **7**, 765-781 (2023). <https://doi.org/10.1016/j.joule.2023.03.005>
46. Journe J M J, Massie W W. Offshore hydromechanics. Delft University of Technology, (2001).
47. Chakrabarti S. Handbook of offshore engineering (2-volume set). Elsevier, (2005).
48. Lawson K W, Lloyd D R. Membrane distillation. *Journal of Membrane Science* **124**, 1-25 (1997). [https://doi.org/10.1016/S0376-7388\(96\)00236-0](https://doi.org/10.1016/S0376-7388(96)00236-0)
49. Balej J. Water vapour partial pressures and water activities in potassium and sodium hydroxide solutions over wide concentration and temperature ranges. *International Journal of Hydrogen Energy* **10**, 233-243 (1985). [https://doi.org/10.1016/0360-3199\(85\)90093-X](https://doi.org/10.1016/0360-3199(85)90093-X)

Comments:

8. Concerning the reproducibility of the research. In the current status the research described in the article is impossible to reproduce, as the current information is not properly organized and it is incomplete. The authors must improve the article to ensure that the research could be reproduced by any other researcher. In addition, some of the problem that this article covers are not properly described. It is not possible for the reader to understand problems like the water migration problem in Figure 1. The electrolysis cell is not properly described, thus, it is not possible to understand their architecture, and how seawater is separated from the SDE. In Figure 2, what is the gas condenser? Is it actually a moisture remover?

Response:

In response to the reviewer's comments, we have provided more details about our seawater electrolysis strategy, the experiment and the specific functions of the floating system in the revised manuscript.

For the **research description**, we separated the laboratory-scale and pilot-scale experiments, and provided a detailed description of the experimental operation process in the "Methods" section of the manuscript.

To explain water migration, we have added a principle description of our seawater electrolysis strategy based on the water phase transition migration mechanism (**Fig. R2_7**, also shown as **Fig. 1a**), and the following sentences and figure have been added to the "Introduction" section (**in line 83-89**) and **Fig. 1a** respectively.

'This strategy is realized by applying a hydrophobic porous membrane as a gas-path interface and employing a concentrated KOH solution as a self-dampening electrolyte (SDE). During operation, the water vapour pressure difference between the seawater and the SDE induces spontaneous seawater gasification on the seawater side and the diffusion of water vapour through the porous membrane to the SDE side, where it is reliquified by absorption by the SDE. When the water migration rate equals the electrolysis rate, continuous and stable water migration via a 'liquid-gas-liquid' mechanism is realized to provide fresh water for electrolysis'

Fig. R2_7. Seawater electrolysis drive by water phase transition migration mechanism.'

For the **gas condenser**, it is a kind of gas-liquid separator (**Fig. R2_8**). The principle of semiconductor refrigeration is adopted to achieve refrigeration and further separate the gas-liquid mixture. In our system, the hydrogen gas (containing alkali steam), generated from an electrolyser, enters the gas condenser through the gas inlet for cooling (1.5 °C), and the alkaline steam is liquified. The cooled and dried hydrogen leaves the condenser through the gas outlet. The description and figure of condenser has been added to the “Methods” section (**in line 420**) and supplementary information:

‘The temperature of electronic condenser (a kind of gas-liquid separator) is set to 1.5 °C, and the setpoint accuracy is ± 0.1 °C.’

Fig. R2_8. Photograph of the condenser.

Comments:

9. About the units in the results. You can provide results in any units you wish but SI units are mandatory. Use MJ for kWh and use kg for Nm³.

Response:

We would like to thank the reviewer very much for the suggestions on physical units. We wrote the units (kWh and Nm³) by following the style of recent literature published by *Nature Communications* (Sun, F. et al. Energy-saving hydrogen production by chlorine-free hybrid seawater splitting coupling hydrazine degradation. *Nat Commun* **12**, 4182 (2021); Shi, H. et al. A sodium-ion-conducted asymmetric electrolyzer to lower the operation voltage for direct seawater electrolysis. *Nat Commun* **14**, 3934 (2023)).

Comments:

10. About the English language: Proof-reading is highly recommended.

Response:

In response to the reviewer’s comment, the manuscript and supplementary information have been carefully proofread. In addition, the language of the entire manuscript has been revised by a professional editor from Springer Nature.

Referee #3

Comments:

1. The manuscript entitled " In-situ direct seawater electrolysis in uncontrollable fluctuating ocean" addressed an important and interesting research study. It is significant to explore to potential opportunities to employ direct seawater electrolysis in the offshore environment. Fluctuating conditions of the ocean are considered in the model development to guide the in-situ direct electrolysis for hydrogen generation in the fluctuating ocean. The methodology is sound and results are noteworthy and support the conclusions drawn from the manuscript, however, further clarification, justification and additional results would help consolidate the idea being presented. Some critical points as follows need to be addressed:

Response:

We would like to thank the reviewer for his/her comments and for summarizing our work. According to the opinions and suggestions of the reviewer, we have revised the manuscript to make the content more solid. Our point-by-point responses to Reviewer 3's comments are as follows.

Comments:

2. Additional details need to be added on sizing of the proposed model. On what basis $1.2 \text{ Nm}^3\text{h}^{-1}$ scale was selected?

Response:

In our study, the size of the floating platform is 7 m (width) \times 9 m (length) \times 2.5 m (height) (the actual space available for layout was 4.4 m (width) \times 6.4 m (length) \times 2.2 m (height), as shown in **Fig. R3_1a**), and the total capacity of seawater electrolysis module was $1.2 \text{ Nm}^3 \text{ H}_2 \text{ h}^{-1}$. The seawater electrolysis module was composed of three parallel electrolyzers, and the scale of each electrolyser was $0.4 \text{ Nm}^3 \text{ H}_2 \text{ h}^{-1}$. The size design of the floating platform was mainly based on the following two considerations:

First, to verify the feasibility of the parallel operation of electrolyzers, we used three electrolyzers (70.5 cm \times 82 cm \times 62 cm) and compared their operating effects in parallel. This could provide a basis for the development of large-scale seawater electrolysis floating platforms with multiple parallel devices in the future.

Second, to verify the impact of a fluctuating environment on seawater electrolysis, the floating platform did not include a propulsion system to avoid interference from other factors; thus, freight equipment had to transport it to the test area, which had many limitations. According to some regulations and restrictions on maritime transportation in China and corresponding safety considerations, only long-distance land transportation from the shipyard to the wharf of the test area, and then a short trip by ship towing to the test area, was allowed. During transportation, limited by road width and height, some road infrastructure such as traffic lights had to be removed (**Fig. R3_1b**). In addition, according to the Chinese Highway Engineering Technical Standards (4), for a secondary

highway, the roadbed width for a design speed of 80 km h⁻¹ was 15.0 m; thus, it was best for the width of the devices to be less than 7.5 m.

Overall, in order to verify the parallel mode of three devices in a limited space, we had to design the floating body to have dimensions of 7 m × 9 m × 2.5 m, with layout dimensions of 4.4 m × 6.4 m × 2.2 m. If the total scale of the electrolyser was further expanded, the floating platform would also be enlarged, which would make it more difficult to transport it on land, so we chose a hydrogen production equipment of 1.2 Nm³ h⁻¹H₂ after comprehensive consideration.

Fig. R3_1. a, Dimensions of the actual layout system inside the floating platform. b, Photographs of the floating platform land transport site.

Comments:

3. A 240-h stable electrolysis operation with an energy consumption of 5 kWh Nm⁻³ H₂. How is this energy consumption is determined and how the components such as seawater electrolysis and energy storage behave in real-time on the fluctuation in wind power?

Response:

We would like to thank the reviewer for his/her comment. The calculation method of electrolytic energy consumption is as follows, which has been added in the ‘Energy consumption associated with H₂ production’ of the Methods section (in line 492-502).

‘The energy consumption for producing hydrogen per Nm³ of H₂ of the three electrolyzers, Q, was calculated as follows:

$$W = I \times \int U dt$$

where I is the current and U is the voltage. W of the three electrolyzers corresponds to W₁, W₂ and W₃. Next, the volume of H₂ produced, V, was calculated as follows:

$$V = 22.4 \times I \times t / (Z \times F)$$

where the number of electrons transferred for the hydrogen evolution reaction, Z, is 2 and the Faraday constant, F, is 96,485 C mol⁻¹. V of the three electrolyzers correspond to V₁, V₂ and V₃.

Thus, the electrolytic energy consumption is Q = W/V (kWh Nm⁻³), and the average electrolytic energy consumption is Q = (W₁+W₂+W₃)/(V₁+V₂+V₃).’

In response to reviewer’s question of how the components behave, the mechanism of the real-time response of energy storage and our seawater electrolyser to fluctuation in wind power is as follows (**Fig. R3_2**): When the offshore wind was sufficient, the offshore wind turbine could be used as the main power supply, and a 220 V±1% stable constant frequency AC is output through the uninterruptible power supply system. Then, the output current was converted into stable DC to supply the seawater electrolysis equipment via a DC-regulated power supply. Due to the fluctuation in offshore wind, the excess electricity was stored in the energy storage device when the wind power was sufficient, while the electricity stored in energy storage device was used to supply for floating platform during a lack of wind power. Since the power of our floating platform was only 19.8 kW, the electricity generated from the 10 MW wind turbine during the offshore test could basically meet the requirements of the platform. Nevertheless, the energy storage device ensured a stable supply of electricity and was able to provide additional power to the platform within 10 ms when the offshore wind became insufficient (**Fig.R3_3**). In light of this, electrolyzers exhibited excellent stability for 10 days under this fluctuating environment, as shown in **Fig. 4** in the manuscript.

Fig. R3_2. Flowchart of a system using fluctuating offshore wind power as an energy source for direct electrolysis of seawater to produce hydrogen.

Fig. R3_3. Stability of the energy storage device.

Comments:

4. Authors assumed the wind speed range of 0-15 m/s. How will the model behave if wind speed increases from the assumed limit?

Response:

We would like to thank the reviewer for his/her comments, which helps further improving our manuscript. In the manuscript, the wind speed range of 0~15 m s⁻¹ was the real date during the offshore test in Xinghua Bay. In this condition, a 10 days stable electrolysis operation with an electrolytic energy consumption of 5 kWh Nm⁻³ H₂ and high-purity of over 99.9% was achieved. This mainly resulted from the fact that the floating platform body resisted most of the wind and wave impacts. In fact, we only had 10 days to carry out the offshore experiment according to the China Meteorological Administration and the Maritime Safety Administration, and we had to evacuate the testing area and return to the dock when the wind level exceeded level 8.

It is necessary to study the influence of stronger wind and waves on our floating platform. Thus, according to the reviewer's suggestion, a finite element simulation has been carried out to analyse the motion characteristics and stress distribution of the floating platform under force 8 wind (about 17.2 m s⁻¹) and the significant wave height of 1 m (it is worth noting that it is dangerous for vessels and platforms to work at sea under force 8 wind). As shown in **Fig. R3_4**, the floating platform is kept stable between 5.5 and 7.9 m in the X direction (sway motion) under the action of the mooring cable. The displacement of the platform in the Y direction (surge motion) is within the range of -0.55 to 0.56 m, and the displacement in the Z direction (heave motion) is within the range of -0.54 to 0.9 m. The floating platform deviates from -2.5 to 2.0° around the X-axis (roll), from -8.5 to 7.2° around the Y-axis (pitch), and from -12.4 to 10.2° around the Z-axis (yaw). As shown in **Fig. R3_5**, when the maximum mooring force on the floating platform is 73.2 kN, the maximum stress occurs at the anchor frame (~286 MPa), and the maximum fluid pressure on the wet surface of the floating platform body is approximately 13041 Pa, which are lower than the yield stress of ship structural steel (355 MPa) we used. Therefore, it is sufficiently safe to carry out our offshore work under 0~0.9 m wave height, 0~15 m s⁻¹ wind speed. Anyway, it is necessary to further optimize the design of a floating platform for industrial applications in marine environments in the future.

The corresponding description has been added in the manuscript (**in line 236-246 and line 349-351**):

*'To ensure the stability and safety of the floating platform in the ocean, the finite element simulation was used to analyse the motion, pressure and stress distribution of the floating platform under a force 8 wind (~ 17.2 m s⁻¹) and a 1-m significant wave height⁴⁶⁻⁴⁷. The floating platform, stabilized by mooring ropes, could swing between 5.5 and 7.9 m along the X direction (sway motion), while it could surge and heave along the Y direction (-0.55~0.56 m) and Z direction (0.54~0.9 m), respectively. The floating platform rotated from -2.5 to 2.0° around the X-axis (roll), from -8.5 to 7.2° around the Y-axis (pitch), and from -12.4 to 10.2° around the Z-axis (yaw), which was within a controllable range (**Supplementary Fig. 9**). In addition, as shown in **Fig. 3c**, maximum stress occurs at the anchor frame (~286 MPa), and the maximum fluid pressure on the wet surface of the floating platform body was approximately 13041 Pa. These values were lower than the yield stress of the ship structural steel employed, ensuring adequate strength for the floating platform.'*

'For use in the deep seas, the resistance of the floating platform to stronger winds and waves should be considered; moreover, the stability and strength of the hydrophobic porous membrane and electrolysis system still need to be optimized for long-term use under controlled ocean conditions.'

Fig. R3_4. Motion distance in the X direction (sway) (a), Y direction (surge) (b) and Z direction (heave) (c). Deviation angle in X direction (roll) (d), Y direction (pitch) (e) and Z direction (yaw) (f).

Fig. R3_5. **a**, The stress distribution of the floating platform. **b**, Hydrodynamic pressure distribution on the wet surface of the floating platform.

Comments:

5. Some insights and at least two cases would be required how the modeled system is producing $1.2 \text{ Nm}^3\text{h}^{-1}$ of hydrogen when the input power source is intermittent in nature, how imperative part is being played by the energy storage system and some details on the storage system sizing would add value.

Response:

We would like to thank the reviewer very much for this comment.

When the offshore wind is sufficient, the wind turbine can generate $220 \text{ V} \pm 1\%$ AC power to the floating plate through an uninterrupted power supply (UPS), and then the DC-regulated power can supply stable DC power to the seawater electrolyzers. Moreover, the excess electricity output by the wind turbine is stored in the energy storage device. When the offshore wind is insufficient to drive the wind turbine's operation, the energy storage serves as the main power source to provide a stable voltage for the floating platform. This design could ensure the stability of the hydrogen production process. The corresponding description has been added to the revised manuscript (**in line 217-224**):

'Fluctuations in the wind velocity cause instability in the power output of wind turbines, which is a challenge for the AWE electrolyzers due to their lower response rate. Accordingly, a UPS with an energy storage device was used to stabilize the output voltage of the turbine. The UPS could stably output 19.8 kW of power, providing continuous energy input for the floating seawater electrolysis platform, while excess electricity was stored in the energy storage device. When the offshore wind was insufficient to drive the wind turbine, the energy storage device served as the main power source, providing a stable voltage for the floating platform. This design could ensure the stability of the hydrogen production process.'

In addition, the energy storage device mainly consists of a system cabinet ($2400 \text{ mm} \times 800 \times 1800 \text{ mm}$) and a battery pack ($4153.9 \text{ mm} \times 600 \text{ mm} \times 1800 \text{ mm}$; rated energy of 188.16 kWh, rated voltage of 224 V, and maximum discharge current of 400 A). The power conversion from the wind turbine to the battery supply could be completed within 10 ms to ensure the continuous normal operation of the floating platform.

A corresponding detailed description of storage energy device has been added to the "**Methods**" section of the manuscript (**in line 393-399**)

'The energy storage device mainly consists of a system cabinet and a battery pack. The system cabinet ($2400 \text{ mm} \times 800 \text{ mm} \times 1800 \text{ mm}$) contains rectifiers, inverters and bypasses. The battery pack ($4153.9 \text{ mm} \times 600 \text{ mm} \times 1800 \text{ mm}$) is composed of a lithium iron phosphate battery, and its rated energy, rated voltage and maximum discharge current are 188.16 kWh, 224 V, and 400 A, respectively. Power conversion from the wind turbine to the battery supply could be completed within 10 ms to ensure the continuous and normal operation of the floating platform.'

Comments:

6. SEM images are shown for the catalysts and PTFE membrane after 10 days seawater electrolysis, however, it would be imperative to see the long term effects on the electrodes, catalysts and membrane to predict the lifetime.

Response:

We sincerely appreciate the reviewer's useful suggestion, which is very important for improving the quality of our paper.

In fact, the seawater electrolysis strategy was based on our previous paper (*Nature* **612**, 673-678 (2022)). In that paper, the system was stably operated for over 3200 h with no obvious electrocatalyst corrosion or membrane wetting was observed, which lays the foundation for long-term use.

Here, a seawater cabin (1.4 m × 1.2 m × 1.0 m) with a wave simulation function (wave height: ~3 cm; wavelength: ~30 cm) was built (**Fig. R3_6a**). After operating for over 500 h in a fluctuating environment, the anode/cathode catalysts and PTFE membrane were observed via SEM (**Fig. R3_6b**), and contact angle was further investigated (**Fig. R3_6c**). The pores of the PTFE membrane were clear, and there was no obvious pore plugging or morphological changes. The catalysts exhibited no obvious corrosion phenomenon, and the catalytic site was clear and uniform. The PTFE membrane still exhibited hydrophobic performance. The experimental results showed that the seawater electrolyser had a durability for over 500 h in a fluctuating environment and had the potential to contribute to long-duration and stable seawater electrolysis in the ocean.

We have added the following description to the manuscript (**in line 138-141**):

'In addition, a 500-h durability test of seawater electrolysis in a simulated fluctuating environment was carried out in the laboratory. The SEM images of the clear catalyst and membrane pores provide evidence of the system's adaptability and robustness in such dynamic settings'

Fig. R3_6. **a**, Photograph of the seawater electrolyser testing in the seawater cabin. **b**, SEM images of the anode catalyst and PTFE membrane before and after the 500-h durability test. **c**, Contact angle of PTFE membrane before and after test. **d**, Stability curve of electrolysis.

REVIEWERS' COMMENTS

Reviewer #1 (Remarks to the Author):

From my point of view, the authors fully answered all my questions. Being satisfied with the answers and the additions made to the manuscript, my opinion is that the work can be published in its current form.

Reviewer #2 (Remarks to the Author):

The Reviewer is satisfied with the changes. Good work.

Reviewer #3 (Remarks to the Author):

The authors of the manuscript entitled "In-situ Direct Seawater Electrolysis Using a Floating Platform in the Ocean with Uncontrollable Wave Motion" have made significant and substantial improvements to the manuscript and addressed all the comments and suggestions accordingly. The revisions made have significantly improved the clarity, coherence, and scientific rigor of the manuscript. Based on the revised content, I am confident in recommending the possible publication of this manuscript.

Point-by-point responses to the reviewers' comments

Referee #1

Comments:

From my point of view, the authors fully answered all my questions.

Being satisfied with the answers and the additions made to the manuscript, my opinion is that the work can be published in its current form.

Response:

We would like to thank the reviewer very much for his/her positive comments and this would significantly encourage us.

Referee #2

Comments:

The Reviewer is satisfied with the changes. Good work.

Response:

We are very grateful to the reviewer for his/her positive evaluation of our work.

Referee #3

Comments:

The authors of the manuscript entitled "In-situ Direct Seawater Electrolysis Using a Floating Platform in the Ocean with Uncontrollable Wave Motion" have made significant and substantial improvements to the manuscript and addressed all the comments and suggestions accordingly. The revisions made have significantly improved the clarity, coherence, and scientific rigor of the manuscript. Based on the revised content, I am confident in recommending the possible publication of this manuscript.

Response:

We would like to thank the reviewer for his/her comments and for summarizing our revised manuscript previously. We also greatly appreciate the specific comments provided by the reviewers in the early stage, which have enabled our work to meet the requirements of *Natural Communications*.